# Minimally distorted Adversarial Examples with a Fast Adaptive Boundary Attack

## Abstract

The evaluation of robustness against adversarial manipulations of neural networks-based classifiers is mainly tested with empirical attacks as the methods for the exact computation, even when available, do not scale to large networks. We propose in this paper a new white-box adversarial attack wrt the $l_p$-norms for $p \in \{1, 2, \infty\}$ aiming at finding the minimal perturbation necessary to change the class of a given input. It has an intuitive geometric meaning, yields quickly high quality results, minimizes the size of the perturbation (so that it returns the robust accuracy at every threshold with a single run). It performs better or similarly to state-of-the-art attacks which are partially specialized to one $l_p$-norm.

## 1 Introduction

The finding of the vulnerability of neural networks-based classifiers to adversarial examples, that is small perturbations of the input able to modify the decision of the models, started a fast development of a variety of attack algorithms. The high effectiveness of adversarial attacks reveals the fragility of these networks which questions their safe and reliable use in the real world, especially in safety critical applications. Many defenses have been proposed to fix this issue (Gu & Rigazio, 2015; Zheng et al., 2016; Papernot et al., 2016; Huang et al., 2016; Bastani et al., 2016; Madry et al., 2018), but with limited success, as new more powerful attacks showed (Carlini & Wagner, 2017b; Athalye et al., 2018; Mosbach et al., 2018). In order to trust the decision of a model, it is necessary to evaluate the exact adversarial robustness. Although this is possible for ReLU networks (Katz et al., 2017; Tjeng et al., 2019) these techniques do not scale to commonly used large networks. Thus, the robustness is evaluated approximating the solution of the minimal adversarial perturbation problem through adversarial attacks.

One can distinguish attacks into black-box (Narodytska & Kasiviswanathan, 2016; Brendel et al., 2018; Su et al., 2019), where one is only allowed to query the classifier, and white-box attacks, where one has full control over the network, according to the attack model used to create adversarial examples (typically some $l_p$-norm, but others have become popular as well, e.g. Brown et al. (2017); Engstrom et al. (2017); Wong et al.), whether they aim at the minimal adversarial perturbation (Carlini & Wagner, 2017a; Chen et al., 2018; Croce et al., 2019) or rather any perturbation below a threshold (Kurakin et al., 2017; Madry et al., 2018; Zheng et al., 2019), if they have lower (Moosavi-Dezfooli et al., 2016; Modas et al., 2019) or higher (Carlini & Wagner, 2017a; Croce et al., 2019) computational cost. Moreover, it is clear that due to the non-convexity of the problem there exists no universally best attack (apart from the exact methods), since this depends on runtime constraints, networks architecture, dataset, etc. However, our goal is to have an attack which performs well under a broad spectrum of conditions with minimal amount of hyperparameter tuning.

In this paper we propose a new white-box attacking scheme which performs comparably or better than established attacks and has the following features: first, it tries to produce adversarial samples with *minimal distortion* compared to the original point, measured wrt the $l_p$-norms with $p \in \{1, 2, \infty\}$. Respect to the quite popular PGD-attack of Madry et al. (2018) this has the clear advantage that our method does not need to be restarted for every threshold $\epsilon$ if one wants to evaluate the success rate of the attack with perturbations constrained to be in $\{\delta \in \mathbb{R}^d \mid \|\delta\|_p \leq \epsilon\}$. Thus it is particularly suitable to get a complete

picture on the robustness of a classifier with low computational cost. Second, it achieves *fast* good quality in terms of average distortion or robust accuracy. At the same time we show that increasing the number of restarts keeps improving the results and makes it competitive with the strongest available attacks. Third, although it comes with a few parameters, these mostly generalize across datasets, architectures and norms considered, so that we have an almost *off-the-shelf method*. Most importantly, unlike PGD and other methods, there is no step size parameter which potentially has to be carefully adapted to every new network.

## 2 FAB: A FAST ADAPTIVE BOUNDARY ATTACK

We first introduce minimal adversarial perturbations, then we recall the definition and properties of the projection wrt the $l_p$-norms of a point on the intersection of a hyperplane and box constraints, as they are an essential part of our attack. Finally, we present our FAB-attack algorithm to generate minimally distorted adversarial examples.

### 2.1 MINIMAL ADVERSARIAL EXAMPLES

Let $f : \mathbb{R}^d \to \mathbb{R}^K$ be a classifier which assigns every input $x \in \mathbb{R}^d$ (with $d$ the dimension of the input space) to one of the $K$ classes according to $\arg\max_{r=1,...,K} f_r(x)$. In many scenarios the input of $f$ has to satisfy a specific set of constraints $C$, e.g. images are represented as elements of $[0, 1]^d$. Then, given a point $x \in \mathbb{R}^d$ with true class $c$, we define the *minimal adversarial perturbation* for $x$ wrt the $l_p$-norm as

$$\delta_{\min,p} = \arg\min_{\delta \in \mathbb{R}^d} \|\delta\|_p, \quad \text{s.th.} \quad \max_{l \neq c} f_l(x + \delta) \geq f_c(x + \delta), \quad x + \delta \in C. \tag{1}$$

The optimization problem (1) is non-convex and NP-hard for non-trivial classifiers (Katz et al. (2017)) and, although for some classes of networks it can be formulated as a mixed-integer program (see Tjeng et al. (2019)), the computational cost of solving it is prohibitive for large, normally trained networks. Thus, $\delta_{\min,p}$ is usually approximated by an *attack algorithm*, which can be seen as a heuristic to solve (1). We will see in the experiments that current attacks sometimes drastically overestimate $\|\delta_{\min,p}\|_p$ and thus the robustness of the networks.

### 2.2 PROJECTION ON A HYPERPLANE WITH BOX CONSTRAINTS

Let $w \in \mathbb{R}^d$ and $b \in \mathbb{R}$ be the normal vector and the offset defining the hyperplane $\pi : \langle w, x \rangle + b = 0$. Let $x \in \mathbb{R}^d$, we denote by the *box-constrained projection* wrt the $l_p$-norm of $x$ on $\pi$ (projection onto the intersection of the box $C = \{z \in \mathbb{R}^d : l_i \leq z_i \leq u_i\}$ and the hyperplane $\pi$) the following minimization problem:

$$z^* = \arg\min_{z \in \mathbb{R}^d} \|z - x\|_p \quad \text{s.th.} \quad \langle w, z \rangle + b = 0, \quad l_i \leq z_i \leq u_i, \quad i = 1, \ldots, d, \tag{2}$$

where $l_i, u_i \in \mathbb{R}$ are lower and upper bounds on each component of $z$. For $p \geq 1$ the optimization problem (2) is convex. Hein & Andriushchenko (2017) proved that for $p \in \{1, 2, \infty\}$ the solution can be obtained in $\mathcal{O}(d \log d)$ time, that is the complexity of sorting a vector of $d$ elements, as well as determining that it has no solution.

Since this projection is part of our iterative scheme, we need to handle specifically the case of (2) being infeasible. In this case, defining $\rho = \text{sign}(\langle w, x \rangle + b)$, we instead compute

$$z' = \arg\min_{z \in \mathbb{R}^d} \rho(\langle w, z \rangle + b) \quad \text{s.th.} \quad l_i \leq z_i \leq u_i, \quad i = 1, \ldots, d, \tag{3}$$

whose solution is given componentwise, for every $i = 1, \ldots, d$, by $z_i = \begin{cases} l_i & \text{if } \rho w_i > 0, \\ u_i & \text{if } \rho w_i < 0, \\ x_i & \text{if } w_i = 0 \end{cases}$.

Assuming that the point $x$ satisfies the box constraints (as it will be in our algorithm), this is equivalent to identifying the corner of the $d$-dimensional box defined by the componentwise constraints on $z$ closest to the hyperplane $\pi$. Notice that if (2) is infeasible then the objective

function of (3) stays positive and the points $x$ and $z$ are strictly contained in the same of the two halfspaces divided by $\pi$. Finally, we define the operator

$$\text{proj}_p : (x, \pi, C) \longmapsto \begin{cases} z^* & \text{if Problem (2) is feasible} \\ z' & \text{else} \end{cases} \tag{4}$$

yielding the point which gets as close as possible to $\pi$ without violating the box constraints.

## 2.3 FAB ATTACK

We introduce now our algorithm to produce minimally distorted adversarial examples, wrt any $l_p$-norm for $p \in \{1, 2, \infty\}$, for a given point $x_{\text{orig}}$ initially correctly classified by $f$ as class $c$. The high-level idea is that we use the linearization of the classifier at the current iterate $x^{(i)}$, compute the box-constrained projections of $x^{(i)}$ respectively $x_{\text{orig}}$ onto the approximated decision hyperplane and take a convex combinations of these projections depending on the distance of $x^{(i)}$ and $x_{\text{orig}}$ to the decision hyperplane, followed by some extrapolation step. We explain below the geometric motivation behind these steps. The attack closest in spirit is DeepFool (Moosavi-Dezfooli et al. (2016)) which is known to be very fast but suffers from low quality. DeepFool just tries to find the decision boundary quickly but has no incentive to provide a solution close to $x_{\text{orig}}$. Our scheme resolves this main problem and, together with the exact projection we use, leads to a principled way to track the decision boundary (the surface where the decision of $f$ changes) *close* to $x_{\text{orig}}$.

If $f$ was a linear classifier then the closest point to $x^{(i)}$ on the decision hyperplane could be found in closed form. Although neural networks are highly non-linear, ReLU networks (neural networks which use ReLU as activation function) are piecewise affine functions and thus locally a linearization of the network is an exact description of the classifier. Let $l \neq c$, then the decision boundary between classes $l$ and $c$ can be locally approximated using a first order Taylor expansion at $x^{(i)}$ by the hyperplane

$$\pi_l(z) : f_l(x^{(i)}) - f_c(x^{(i)}) + \left\langle \nabla f_l(x^{(i)}) - \nabla f_c(x^{(i)}), z - x^{(i)} \right\rangle = 0. \tag{5}$$

Moreover the $l_p$-distance $d_p(\pi, x^{(i)})$ of $x^{(i)}$ to $\pi_l$ is given by

$$d_p(\pi_l, x^{(i)}) = \frac{|f_l(x^{(i)}) - f_c(x^{(i)})|}{\left\| \nabla f_l(x^{(i)}) - \nabla f_c(x^{(i)}) \right\|_q}, \quad \text{with} \quad \frac{1}{p} + \frac{1}{q} = 1. \tag{6}$$

Note that if $d_p(\pi_l, x^{(i)}) = 0$ then $x^{(i)}$ belongs to the true decision boundary. Moreover, if the local linear approximation of the network is correct then the class $s$ with the decision hyperplane closest to the point $x^{(i)}$ can be computed as

$$s = \arg\min_{l \neq c} \frac{|f_l(x^{(i)}) - f_c(x^{(i)})|}{\left\| \nabla f_l(x^{(i)}) - \nabla f_c(x^{(i)}) \right\|_q}. \tag{7}$$

Thus, given that the approximation holds in some large enough neighborhood, the projection $\text{proj}_p(x^{(i)}, \pi_s, C)$ of $x^{(i)}$ onto $\pi_s$ lies on the decision boundary (unless (2) is infeasible).

**Biased gradient step:** The iterative algorithm $x^{(i+1)} = \text{proj}_p(x^{(i)}, \pi_s, C)$ would be similar to DeepFool except that our projection operator is exact whereas they project onto the hyperplane and then clip to $[0, 1]^d$. This scheme is not biased towards the original target point $x_{\text{orig}}$, thus it goes typically further than necessary to find a point on the decision boundary as basically the algorithm does not aim at the minimal adversarial perturbation. Thus we consider additionally $\text{proj}_p(x_{\text{orig}}, \pi_s, C)$ and use instead the iterative step, with $x^{(0)} = x_{\text{orig}}$, defined as

$$x^{(i+1)} = (1 - \alpha) \cdot \text{proj}_p(x^{(i)}, \pi_s, C) + \alpha \cdot \text{proj}_p(x_{\text{orig}}, \pi_s, C), \tag{8}$$

which biases the step towards $x_{\text{orig}}$ (see Figure 1). Note that this is a convex combination of two points on $\pi_s$ and in $C$ and thus also $x^{(i+1)}$ lies on $\pi_s$ and is contained in $C$. As we wish

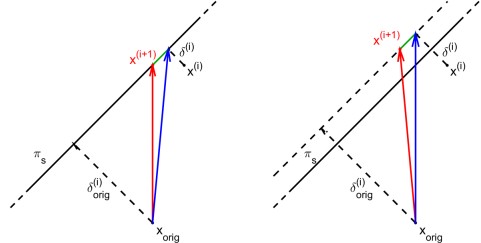

Figure 1: Visualization of FAB-attack scheme, with on the left the case $\eta = 1$, on the right $\eta > 1$. In blue we represent the next iterate $x^{(i+1)}$ one would get without any bias toward the original point $x_{\text{orig}}$, in green the effect of the bias we introduce and in red the $x^{(i+1)}$ obtained with our scheme in (10). We see that our algorithm tends to stay closer to the original point compared to the one with an unbiased gradient step.

a scheme with minimal amount of parameters, we want to have an automatic selection of $\alpha$ based on the available geometric quantities. Let

$$\delta^{(i)} = \text{proj}_p(x^{(i)}, \pi_s, C) - x^{(i)} \quad \text{and} \quad \delta^{(i)}_{\text{orig}} = \text{proj}_p(x_{\text{orig}}, \pi_s, C) - x_{\text{orig}}.$$

Note that $\left\|\delta^{(i)}\right\|_p$ and $\left\|\delta^{(i)}_{\text{orig}}\right\|_p$ are the distances of $x^{(i)}$ and $x_{\text{orig}}$ to $\pi_s$ (inside $C$). We propose to use for the parameter $\alpha$ the relative magnitude of these two distances, that is

$$\alpha = \min \left\{ \frac{\left\|\delta^{(i)}\right\|_p}{\left\|\delta^{(i)}\right\|_p + \left\|\delta^{(i)}_{\text{orig}}\right\|_p}, \alpha_{\max} \right\} \in [0, 1]. \tag{9}$$

The motivation for doing so is that if $x^{(i)}$ is close to the decision boundary, then we should stay close to this point (note that $\pi_s$ is the approximation of $f$ computed at $x^{(i)}$ and thus it is valid in a small neighborhood of $x^{(i)}$, whereas $x_{\text{orig}}$ is farther away). On the other hand we want to have the bias towards $x_{\text{orig}}$ in order not to go too far away from $x_{\text{orig}}$. This is why $\alpha$ depends on the distances of $x^{(i)}$ and $x_{\text{orig}}$ to $\pi_s$ but we limit it from above with $\alpha_{\max}$. Finally, we use a small extrapolation step as we noted empirically, similarly to Moosavi-Dezfooli et al. (2016), that this helps to cross faster the decision boundary and get an adversarial sample. This leads to the final scheme:

$$x^{(i+1)} = \text{proj}_C\left((1 - \alpha)(x^{(i)} + \eta\delta^{(i)}) + \alpha(x_{\text{orig}} + \eta\delta^{(i)}_{\text{orig}})\right), \tag{10}$$

where $\alpha$ is chosen as in (9), $\eta \geq 1$ and $\text{proj}_C$ is just the projection onto the box which can be done by clipping. In Figure 1 we visualize the scheme: in black one can see the hyperplane $\pi_s$ and the vectors $\delta^{(i)}_{\text{orig}}$ and $\delta^{(i)}$, in blue the step we would make going to the decision boundary with the DeepFool variant, while in red the actual step we have in our method. The green vector represents instead the bias towards the original point we introduce. On the left of Figure 1 we use $\eta = 1$, while on the right we use overshooting $\eta > 1$.

**Interpretation of $\text{proj}_p(x_{\text{orig}}, \pi_s, C)$:** The projection of the target point onto the intersection of $\pi_s$ and $C$ is defined as

$$\underset{z \in \mathbb{R}^d}{\arg\min} \ \|z - x_{\text{orig}}\|_p \quad \text{s.th.} \quad \langle w, z \rangle + b = 0, \quad l_i \leq z_i \leq u_i,$$

Note that replacing $z$ by $x^{(i)} + \delta$ we can rewrite this as

$$\underset{\delta \in \mathbb{R}^d}{\arg\min} \ \left\|x^{(i)} + \delta - x_{\text{orig}}\right\|_p \quad \text{s.th.} \quad \langle w, x + \delta \rangle + b = 0, \quad l_i \leq x_i + \delta_i \leq u_i.$$

This can be interpreted as the minimization of the distance of the next iterate $x^{(i)} + \delta$ to the target point $x_{\text{orig}}$ so that $x^{(i)} + \delta$ lies on the intersection of the (approximate) decision hyperplane and the box $C$. This point of view on the projection $\text{proj}_p(x_{\text{orig}}, \pi_s, C)$ again justifies using a convex combination of the two projections in our iterative scheme in (10).

**Backward step:**  The described scheme finds in a few iterations adversarial perturbations. However, we are interested in minimizing their norms. Thus, once we have a new point $x^{(i+1)}$, we check whether it is assigned by $f$ to a class different from $c$. In this case, we apply

$$x^{(i+1)} = (1 - \beta)x_{\text{orig}} + \beta x^{(i+1)}, \quad \beta \in (0, 1), \tag{11}$$

that is we go back towards $x_{\text{orig}}$ on the segment $[x^{(i+1)}, x_{\text{orig}}]$, effectively starting again the algorithm at a point which is quite close to the decision boundary. In this way, due to the bias of the method towards $x_{\text{orig}}$ we successively find adversarial perturbations of smaller norm, meaning that the algorithm *tracks* the decision boundary while getting closer to $x_{\text{orig}}$.

**Final search:**  Our scheme finds points close to the decision boundary but often they are slightly off as the linear approximation is not exact and we apply the extrapolation step with $\eta > 1$. Thus, after finishing $N_{\text{iter}}$ iterations of our algorithmic scheme, we perform a last, fast step to further improve the quality of the adversarial examples. Let $x_{\text{out}}$ be the closest point to $x_{\text{orig}}$ classified differently from $c$, say $s \neq c$, found with the iterative scheme. It holds that $f_s(x_{\text{out}}) - f_c(x_{\text{out}}) > 0$ and $f_s(x_{\text{orig}}) - f_c(x_{\text{orig}}) < 0$. This means that, assuming $f$ continuous, there exists a point $x^*$ on the segment $[x_{\text{out}}, x_{\text{orig}}]$ such that $f_s(x^*) - f_c(x^*) = 0$ and $\|x^* - x_{\text{orig}}\|_p < \|x_{\text{out}} - x_{\text{orig}}\|_p$. If $f$ is linear

$$x^* = x_{\text{out}} - \frac{f_s(x_{\text{out}}) - f_c(x_{\text{out}})}{f_s(x_{\text{out}}) - f_c(x_{\text{out}}) + f_s(x_{\text{orig}}) - f_c(x_{\text{orig}})}(x_{\text{out}} - x_{\text{orig}}). \tag{12}$$

Since $f$ is typically non-linear, but close to linear, we compute iteratively for a few steps

$$x_{\text{temp}} = x_{\text{out}} - \frac{f_s(x_{\text{out}}) - f_c(x_{\text{out}})}{f_s(x_{\text{out}}) - f_c(x_{\text{out}}) + f_s(x_{\text{orig}}) - f_c(x_{\text{orig}})}(x_{\text{out}} - x_{\text{orig}}), \tag{13}$$

each time replacing in (13) $x_{\text{out}}$ with $x_{\text{temp}}$ if $f_s(x_{\text{temp}}) - f_c(x_{\text{temp}}) > 0$ or $x_{\text{orig}}$ with $x_{\text{temp}}$ if instead $f_s(x_{\text{temp}}) - f_c(x_{\text{temp}}) < 0$. With this kind of modified binary search one can find a better adversarial sample with the cost of a few forward passes of the network.

**Random restarts:**  So far all the steps are deterministic. To improve the results, we introduce the option of random restarts, that is $x^{(0)}$ is randomly sampled in proximity of $x_{\text{orig}}$ instead of being $x_{\text{orig}}$ itself. Most attacks benefit from random restarts, e.g. Madry et al. (2018); Zheng et al. (2019), especially dealing with gradient-masking defenses (Mosbach et al. (2018)), as it allows a wider exploration of the input space. We choose to sample from the $l_p$-sphere centered in the original point with radius half the $l_p$-norm of the current best adversarial perturbation (or a given threshold if no adversarial example has been found yet).

**Computational cost:**  Our attack, in Algorithm 1, consists of two main operations: the computation of $f$ and its gradients and solving the projection (2). We perform, for each iteration, a forward and a backward pass of the network in the gradient step and a forward pass in the backward step. The projection can be efficiently implemented to run in batches on the GPU and its complexity depends only on the input dimension. Thus, except for shallow models, its cost is much smaller than the passes through the network. We can approximate the computational cost of our algorithm by the total number of calls of the classifier

$$N_{\text{iter}} \times N_{\text{restarts}} \times (2 \times \text{forward passes} + 1 \times \text{backward pass}). \tag{14}$$

One has to add the forward passes for the final search, fixed to 3, that happens just once.

## 2.4  Comparison to DeepFool

The idea of exploiting the first order local approximation of the decision boundary is not novel but the basis of one of the first white-box adversarial attacks, DeepFool (DF) from Moosavi-Dezfooli et al. (2016). While DF and our FAB-attack share the strategy of using a linear approximation of the classifier and projecting on the decision hyperplanes, we want to point out many key differences: first, DF does not solve the projection (2) but its simpler version without box constraints, clipping afterwards. Second, their gradient step does not have any bias towards the original point, that is equivalent to $\alpha = 0$ in (10). Third, DF does

---

**Algorithm 1:** FAB-attack

**Input** : $x_{\text{orig}}$ original point, $c$ original class, $N_{\text{restarts}}, N_{\text{iter}}, \alpha_{\max}, \beta, \eta, \epsilon, p$
**Output:** $x_{\text{out}}$ adversarial example

**1** $u \leftarrow +\infty$
**2** **for** $j = 1, \ldots, N_{restarts}$ **do**
**3**     **if** $j = 1$ **then** $x^{(0)} \leftarrow x_{\text{orig}}$;
**4**     **else** $x^{(0)} \leftarrow$ randomly sampled s.th. $\left\| x^{(0)} - x_{\text{orig}} \right\|_p = \min\{u,\epsilon\}/2$;
**5**     **for** $i = 0, \ldots, N_{iter} - 1$ **do**
**6**         $s \leftarrow \underset{l \neq c}{\arg\min} \frac{|f_l(x^{(i)}) - f_c(x^{(i)})|}{\left\| \nabla f_l(x^{(i)}) - \nabla f_c(x^{(i)}) \right\|_q}$
**7**         $\delta^{(i)} \leftarrow \text{proj}_p(x^{(i)}, \pi_s, C)$
**8**         $\delta_{\text{orig}}^{(i)} \leftarrow \text{proj}_p(x_{\text{orig}}, \pi_s, C)$
**9**         compute $\alpha$ as in Equation (9)
**10**         $x^{(i+1)} \leftarrow \text{proj}_C\Big((1-\alpha)\left(x^{(i)} + \eta\delta^{(i)}\right) + \alpha(x_{\text{orig}} + \eta\delta_{\text{orig}}^{(i)})\Big)$
**11**         **if** $x^{(i+1)}$ *is not classified in $c$* **then**
**12**             **if** $\left\| x^{(i+1)} - x_{orig} \right\|_p < u$ **then**
**13**                 $x_{\text{out}} \leftarrow x^{(i+1)}$
**14**                 $u \leftarrow \left\| x^{(i+1)} - x_{\text{orig}} \right\|_p$
**15**             **end**
**16**             $x^{(i+1)} \leftarrow (1-\beta)x_{\text{orig}} + \beta x^{(i+1)}$
**17**         **end**
**18**     **end**
**19** **end**
**20** perform 3 steps of final search on $x_{\text{out}}$ as in (13)

---

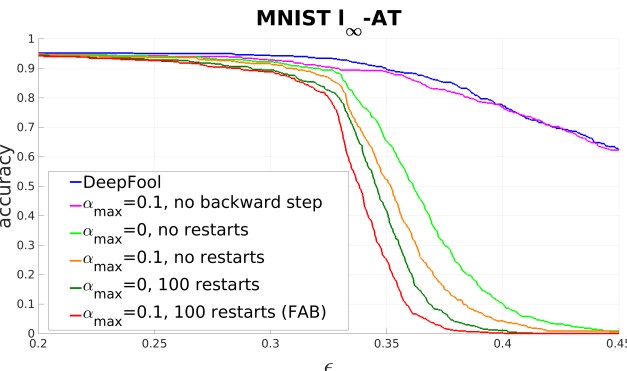

Figure 2: **Ablation study to DeepFool for $l_\infty$-attacks.** The introduction of the convex combination ($\alpha_{\max} = 0.1$, no backward step) already improves over DeepFool. Moreover, if one does our full approach, the case $\alpha_{\max} = 0$ (can be seen as an improved iterative DeepFool) is worse than $\alpha_{\max} = 0.1$ with the same number of restarts. In the plots we show the robust accuracy as a function of the threshold $\epsilon$ under the different attacks on the $l_\infty$-AT model on MNIST.

not have any backward step, final search or restart, as it stops as soon as a misclassified point is found (its goal is to provide quickly an adversarial perturbation of average quality). We perform an ablation study of the differences to DF in Figure 2, where we show the curves of the robust accuracy as a function of the threshold $\epsilon$ (lower is better). We present the results of DeepFool (blue) and FAB-attack with the following variations: $\alpha_{\max} = 0.1$ and no backward step (magenta), $\alpha_{\max} = 0$ (that is no bias in the gradient step) and no restarts (light green), $\alpha_{\max} = 0.1$ and no restarts (orange), $\alpha_{\max} = 0$ and 100 restarts (dark

green) and $\alpha_{\max} = 0.1$ and 100 restarts, that is FAB-attack, (red). We can see how every addition we make to the original scheme of DeepFool contributes to the significantly improved performance of FAB-attack when compared to the original DeepFool.

## 3 EXPERIMENTS

**Models:** We run experiments on MNIST, CIFAR-10 (Krizhevsky et al.) and Restricted ImageNet (Tsipras et al. (2019)). For each dataset we consider a naturally trained model (*plain*) and two adversarially trained ones as in Madry et al. (2018), one to achieve robustness wrt the $l_\infty$-norm ($l_\infty$-AT) and the other wrt the $l_2$-norm ($l_2$-AT) (see A.1).

**Attacks:** We compare the performances of FAB-attack to those of attacks representing the state-of-the-art in each norm: DeepFool (DF) (Moosavi-Dezfooli et al. (2016)), Carlini-Wagner $l_2$-attack (CW) (Carlini & Wagner (2017a)), Linear Region $l_2$-Attack (LRA) (Croce et al. (2019)), Projected Gradient Descent on the cross-entropy function (PGD) (Kurakin et al., 2017; Madry et al., 2018; Tramèr & Boneh, 2019), Distributionally Adversarial Attack (DAA) (Zheng et al. (2019)), SparseFool (SF) (Modas et al. (2019)), Elastic-net Attack (EAD) (Chen et al. (2018)). We use DF from Rauber et al. (2017), CW and EAD as in Papernot et al. (2017), DAA and LRA with the code from the original papers, while we reimplemented SF and PGD. For MNIST and CIFAR-10 we used DAA with 50 restarts, PGD and FAB with 100 restarts. For Restricted ImageNet, we used DAA, PGD and FAB with 10 restarts (for $l_1$ we used 5 restarts, since both methods benefit from more iterations). Moreover, we could not use LRA since it hardly scales to such models and CW and EAD for compatibility issues between the implementations of attacks and models. See A.2 for more details e.g. regarding number of iterations and other hyperparameters.

**Evaluation metrics:** The *robust accuracy* of a model at a threshold $\epsilon$ is defined as the classification accuracy (in percentage) the model achieves when an attack is allowed to change every input of the test set with perturbations of $l_p$-norm smaller than $\epsilon$ in order to change the decision. Thus stronger attacks produce lower robust accuracies. For each model and dataset we fix five thresholds at which we compute the robust accuracy for each attack (we choose the thresholds to have values of the robust accuracy that cover the range between clean accuracy and 0). We evaluate the attacks through the following statistics: i) **avg. rob. accuracy**: the mean of all the values of robust accuracy given by the attack over all models and thresholds, ii) **# best**: how many times the attack achieves the lowest robust accuracy (it is the most effective), iii) **avg. difference to best**: for each model/threshold we compute the difference between the robust accuracy of the attack and the best one across all the attacks, then we average over all models/thresholds, iv) **max difference to best**: as "avg. difference to best", but with the maximum difference instead of the average one. In A.4 we report the average $l_p$-norm of the adversarial perturbations given by the attacks.

**Results:** We report the complete results in Tables 5 to 13 of the Appendix, while we summarize them in Tables 1 (MNIST and CIFAR-10 aggregated, as we used the same attacks) and 2 (Restricted ImageNet). Our FAB-attack achieves the best results in all statistics for every norm (with the only exception of "max diff. to best" in $l_\infty$) on MNIST+CIFAR-10, meaning that it is the most effective attack. In particular, while on $l_\infty$ the "avg. robust accuracy" of PGD is not far from that of FAB, the gap is large when considering $l_2$ and $l_1$. Interestingly, the second best attack, at least in terms of average robust accuracy, is different for every norm (PGD for $l_\infty$, LRA for $l_2$, EAD for $l_1$), which implies that FAB outperforms algorithms specialized in the individual norms.

We also report the results of FAB-10, that is our attack with only 10 restarts, to show that FAB yields high quality results already with a low budget in terms of time/computational cost. In fact, FAB-10 has "avg. robust accuracy" better than or very close to that of the strongest versions of the other methods (see below for a runtime analysis, where one observes that FAB-10 is the fastest attack excluding DF and SF which however give much worse results). On Restricted ImageNet, FAB-attack gets the best results in all statistics for $l_1$, while for $l_\infty$ and $l_2$, although PGD performs often better, the difference in "avg. robust accuracy" is small, meaning that FAB performs mostly similarly to PGD.

Table 1: Performance summary of all attacks on MNIST and CIFAR-10 (aggregated). We report, for each norm, "avg. rob. acc.", the mean of the values of robust accuracy across all the models and datasets, "# best", number of times the attack is the best one, "avg. diff. to best" and "max diff. to best", the mean and maximum differences between the robust accuracy of the attack and that of the best attack for each model/threshold (on the first 1000 points for $l_\infty$ and $l_1$, 500 for $l_2$, of the test sets). The numbers after the name of the attacks indicates the number of restarts used. In total we consider 5 thresholds $\times$ 6 models = 30 cases for each of the 3 norms. *Note that for FAB-10 (i.e. with 10 restarts) the "# best" is computed excluding the results of FAB-100.

**statistics on MNIST + CIFAR-10**

| $l_\infty$-norm | | | DF | DAA-50 | PGD-100 | FAB-10 | FAB-100 |
|---|---|---|---|---|---|---|---|
| avg. rob. acc. | | | 58.81 | 60.67 | 46.07 | 46.18 | **45.47** |
| # best | | | 0 | 8 | 12 | 13* | **17** |
| avg. diff. to best | | | 14.58 | 16.45 | 1.85 | 1.96 | **1.25** |
| max diff. to best | | | 78.10 | 49.00 | **10.70** | 20.30 | 17.10 |
| | | | | | | | |
| $l_2$-norm | CW | DF | LRA | PGD-100 | FAB-10 | FAB-100 |
| avg. rob. acc. | 45.09 | 56.10 | 36.97 | 44.94 | 36.41 | **35.57** |
| # best | 4 | 1 | 9 | 11 | 19* | **23** |
| avg. diff. to best | 9.65 | 20.67 | 1.54 | 9.51 | 0.98 | **0.13** |
| max diff. to best | 65.40 | 91.40 | 13.60 | 64.80 | 8.40 | **1.60** |
| | | | | | | |
| $l_2$-norm wo/ Madry's model | CW | DF | LRA | PGD-100 | FAB-10 | FAB-100 |
| avg. rob. acc. | 40.85 | 49.18 | 40.25 | 42.95 | 39.98 | **39.57** |
| # best | 4 | 1 | 8 | 11 | 14* | **18** |
| avg. diff. to best | 1.44 | 9.78 | 0.84 | 3.54 | 0.57 | **0.16** |
| max diff. to best | 8.80 | 44.00 | 4.00 | 22.00 | 4.20 | **1.60** |
| | | | | | | |
| $l_1$-norm | | | SF | EAD | PGD-100 | FAB-10 | FAB-100 |
| avg. rob. acc. | | | 64.47 | 35.79 | 49.51 | 33.26 | **29.46** |
| # best | | | 0 | 13 | 0 | 10* | **17** |
| avg. diff. to best | | | 35.31 | 6.63 | 20.35 | 4.10 | **0.30** |
| max diff. to best | | | 95.90 | 58.40 | 74.00 | 21.80 | **1.60** |
| | | | | | | | |
| $l_1$-norm wo/ Madry's model | | | SF | EAD | PGD-100 | FAB-10 | FAB-100 |
| avg. rob. acc. | | | 58.06 | 32.56 | 43.82 | 33.79 | **32.06** |
| # best | | | 0 | **13** | 0 | 5* | 12 |
| avg. diff. to best | | | 26.36 | 0.87 | 12.12 | 2.10 | **0.36** |
| max diff. to best | | | 53.10 | 4.80 | 31.90 | 3.90 | **1.60** |

In general, both average and maximum *difference to best* of FAB-attack are small for all the datasets and norms, implying that it does not suffer severe failures, which makes it an efficient, high quality technique to evaluate the robustness of classifiers for all $l_p$-norms. Finally, we show in Table 4 that FAB-attack outperforms or matches the competitors in 16 out of 18 cases when comparing the average $l_p$-norms of the generated adversarial perturbations.

**Runtime comparison:** DF and SF are definitely much faster than the others as their primary goal is to find as soon as possible adversarial examples, without emphasis on minimizing their norms, while LRA is rather expensive as noted in the original paper. Below we report the runtimes (for 1000 points on MNIST and CIFAR-10, 50 on R-ImageNet) for the attacks as used in the experiments (if not specified otherwise, it includes all the restarts). For PGD and DAA this is the time for evaluating the robust accuracy at 5 thresholds, while for the other methods a single run is sufficient to compute all the statistics.
**MNIST**: DAA-50 11736s, PGD-100 3825s for $l_\infty/l_2$ and PGD-100 14106s for $l_1$, CW 944s, EAD 606s, FAB-10 161s, FAB-100 1613s. **CIFAR-10**: DAA-50 11625s, PGD-100 31900s

Table 2: As in Table 1 statistics of the performance of different attacks on Restricted ImageNet (on the first 500 points of the validation set). In total we consider 5 thresholds $\times$ 3 models = 15 cases for each of the 3 norms.

**statistics on Restricted ImageNet**

|  | $l_\infty$-norm | | | | $l_2$-norm | | | $l_1$-norm | | |
|---|---|---|---|---|---|---|---|---|---|---|
|  | DF | DAA-10 | PGD-10 | FAB-10 | DF | PGD-10 | FAB-10 | EAD | PGD-5 | FAB-5 |
| avg. rob. acc. | 35.61 | 38.44 | **26.91** | 27.83 | 45.69 | **31.75** | 33.24 | 71.31 | 40.64 | **38.12** |
| # best | 0 | 1 | **13** | 3 | 0 | **14** | 1 | 0 | 3 | **12** |
| avg. diff. best | 8.75 | 11.57 | **0.04** | 0.96 | 13.99 | **0.04** | 1.53 | 33.52 | 2.85 | **0.33** |
| max diff. best | 14.60 | 37.20 | **0.40** | 2.00 | 25.40 | **0.60** | 3.40 | 59.00 | 6.20 | **2.40** |

for $l_\infty/l_2$ and 70110s for $l_1$, CW 3691s, EAD 3398s, FAB-10 1209s, FAB-100 12093s. **R-ImageNet**: DAA-10 6890s, PGD-10 4738s for $l_\infty/l_2$ and PGD-5 24158s for $l_1$, FAB-10 2268s for $l_\infty/l_2$ and FAC-5 3146s for $l_1$ (note that different numbers of restarts/iterations for $l_1$ are used on R-ImageNet).

PGD needs a forward and a backward pass of the network for each iteration. Thus it is given 1.5 times more iterations than FAB, so that overall they have same budget of passes (we assume here that forward and backward passes take the same amount of time). In Appendix B.2 we compare PGD-1 versus FAB-1 as a function of the number of passes (2 passes are one iteration of PGD, 3 passes are one iteration of FAB - the plots show 300 passes meaning 150 iterations of PGD and 100 iterations of FAB) so that the comparison is fair concerning runtime in order to compare the performance of PGD-1 and FAB-1 as a function of runtime. If one considers just the performance up to 20 passes (10 iterations PGD, 7 iterations FAB) then FAB outperforms PGD in 18 out of 27 cases. However, one also observes that there is no general superiority of one method. For both PGD-1 and FAB-1 there are cases where the method requires the full amount of 300 passes to get to a good performance whereas the other method achieves it with significantly less iterations/passes.

## 4 CONCLUSION

In summary, our geometrically motivated FAB-attack outperforms in terms of runtime and on average in terms of quality all other high quality state-of-the-art attacks and can be used for all $p$-norms in $p \in \{1, 2, \infty\}$ which is not the case for most other methods.

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

# A  EXPERIMENTS

## A.1  MODELS

The *plain* and $l_\infty$-AT models on MNIST are those available at `https://github.com/MadryLab/mnist_challenge` and consist of two convolutional and two fully-connected layers. The architecture of the CIFAR-10 models has 8 convolutional layers (with number of filters increasing from 96 to 384) and 2 dense layers, while on Restricted ImageNet we use the models (ResNet-50 He et al. (2016)) from Tsipras et al. (2019) and available at `https://github.com/MadryLab/robust-features-code`.
The models on MNIST achieve the following clean accuracy: *plain* 98.7%, $l_\infty$-AT 98.5%, $l_2$-AT 98.6%. The models on CIFAR-10 achieve the following clean accuracy: *plain* 89.2%, $l_\infty$-AT 79.4%, $l_2$-AT 81.2%.

## A.2  ATTACKS

We use CW with 10000 iterations and confidence 0, EAD with 1000 iterations, $l_1$ decision rule and $\beta = 0.05$. In both cases we set the parameters to achieve minimally (wrt $l_2$ for CW and $l_1$ for EAD) distorted adversarial examples. We could not use these methods on Restricted ImageNet since, to be compatible with the attack from Papernot et al. (2017), it would be necessary to reimplement from scratch the models of Tsipras et al. (2019), as done in `https://github.com/tensorflow/cleverhans/tree/master/cleverhans/model_zoo/madry_lab_challenges` for a similar situation.

For DAA we use 200 iterations for MNIST, 50 for the other datasets and, given a threshold $\epsilon$, a step size of $\epsilon/30$ for MNIST, $\epsilon/10$ otherwise.

We perform PGD with 150 iterations, except for the case of $l_1$ on Restricted ImageNet where we use 450 iterations. For PGD wrt $l_\infty$ we use, given a threshold $\epsilon$, a step size of $\epsilon/10$ in the direction of the sign of the gradient of the cross entropy loss, for PGD wrt $l_2$ we perform at each iteration a step in the direction of the gradient of size $\epsilon/4$, for PGD wrt $l_1$ we use the gradient step suggested in Tramèr & Boneh (2019) (with sparsity levels of 1% for MNIST and 10% for CIFAR-10 and Restricted ImageNet), with size $\epsilon/2$. The above stepsize parameters $\epsilon/4$ for $l_2$ and $\epsilon/10$ for $l_\infty$ for PGD were obtained, by doing a grid search for each norm separately and using the values working best on average on MNIST and CIFAR-10. In Appendix we discuss the influence of the step-size and show that our chosen values perform best on average, see Figure 3.

For FAB-attack we set 100 iterations, except for the case of $l_1$ on Restricted ImageNet where we use 300 iterations. Moreover, we use the following parameters for all the cases on MNIST and CIFAR-10: $\alpha_{\max} = 0.1$, $\eta = 1.05$, $\beta = 0.9$. On Restricted ImageNet we set $\alpha_{\max} = 0.05$, $\eta = 1.3$, $\beta = 0.9$. When using random restarts, FAB-attack needs a value for the parameters $\epsilon$. It represents the radius of the $l_p$-ball around the original point inside which we sample the starting point of the algorithm, at least until a sufficiently small adversarial perturbation is found (see Algorithm 1). We use the values of $\epsilon$ reported in Table 3. Note however that the attack usually finds at the first run an adversarial perturbation small enough so that $\epsilon$ in practice rarely comes into play.

Table 3: We report the values of $\epsilon$ used for sampling in case our FAB-attack uses random restarts.

**values of $\epsilon$ used for random restarts**

|  | MNIST | | | CIFAR-10 | | | Restricted ImageNet | | |
|---|---|---|---|---|---|---|---|---|---|
|  | plain | $l_\infty$-AT | $l_2$-AT | plain | $l_\infty$-AT | $l_2$-AT | plain | $l_\infty$-AT | $l_2$-AT |
| $l_\infty$ | 0.15 | 0.3 | 0.3 | 0.0 | 0.02 | 0.02 | 0.02 | 0.08 | 0.08 |
| $l_2$ | 2.0 | 2.0 | 2.0 | 0.5 | 4.0 | 4.0 | 5.0 | 5.0 | 5.0 |
| $l_1$ | 40.0 | 40.0 | 40.0 | 10.0 | 10.0 | 10.0 | 100.0 | 250.0 | 250.0 |

### A.3    COMPLETE RESULTS

In Tables 5 to 13 we report the complete values of the robust accuracy, wrt either $l_\infty$, $l_2$ or $l_1$, computed by every attack, for 3 datasets, 3 models for each dataset, 5 thresholds for each model (135 evaluations overall).

### A.4    FURTHER RESULTS

In Table 4 we report the average $l_p$-norm of the adversarial perturbations found by the different attacks, computed on the originally correctly classified points on which the attack is successful. Note that we cannot show this statistic for the attacks which do not minimize the distance of the adversarial example to the clean input (PGD and DAA). FAB-attack produces also in this metric the best results in most of the cases, being the best for every model when considering $l_\infty$ and $l_2$, and the best in 4 out of 6 cases in $l_1$ (lower values mean a stronger attack).

Table 4: We report mean $l_p$-norm of the adversarial perturbations found by the attacks (when successful, excluding the already misclassified points) for every model.

**average norm of adversarial perturbations**

| $l_\infty$-norm | | | | | DF | FAB |
|---|---|---|---|---|---|---|
| | plain | | | | 0.078 | **0.066** |
| MNIST | $l_\infty$-at | | | | 0.508 | **0.326** |
| | $l_2$-at | | | | 0.249 | **0.170** |
| | plain | | | | 0.008 | **0.006** |
| CIFAR-10 | $l_\infty$-at | | | | 0.032 | **0.024** |
| | $l_2$-at | | | | 0.026 | **0.019** |

| $l_2$-norm | | DF | CW | LRA | FAB |
|---|---|---|---|---|---|
| | plain | 1.13 | 1.01 | **1.00** | **1.00** |
| MNIST | $l_\infty$-at | 4.95 | 1.76 | 1.25 | **1.12** |
| | $l_2$-at | 3.10 | 2.35 | 2.25 | **2.24** |
| | plain | 0.28 | **0.21** | 0.22 | **0.21** |
| CIFAR-10 | $l_\infty$-at | 0.96 | 0.74 | 0.74 | **0.73** |
| | $l_2$-at | 0.91 | 0.71 | 0.72 | **0.70** |

| $l_1$-norm | | | | | EAD | FAB |
|---|---|---|---|---|---|---|
| | plain | | | | 6.38 | **6.04** |
| MNIST | $l_\infty$-at | | | | 8.26 | **3.36** |
| | $l_2$-at | | | | 12.18 | **12.16** |
| | plain | | | | 3.01 | **2.87** |
| CIFAR-10 | $l_\infty$-at | | | | **5.79** | 6.03 |
| | $l_2$-at | | | | **7.94** | 8.05 |

Table 5: Comparison of $l_\infty$-, $l_2$- and $l_1$-attacks on a naturally trained model on MNIST. We report the accuracy in percentage of the classifier on the test set if the attack is allowed to perturb the test points of $\epsilon$ in $l_p$-distance. The statistics are computed on the first 1000 points on the test set for $l_\infty$ and $l_1$, on 500 points for $l_2$.

**Robust accuracy of MNIST plain model**

| metric | $\epsilon$ | DF | DAA-1 | DAA-50 | PGD-1 | PGD-10 | PGD-100 | FAB-1 | FAB-10 | FAB-100 |
|---|---|---|---|---|---|---|---|---|---|---|
| | 0.03 | 93.2 | **91.9** | **91.9** | 92.0 | **91.9** | **91.9** | 92.0 | 92.0 | 92.0 |
| | 0.05 | 83.4 | 78.2 | 76.7 | 76.0 | 74.9 | **74.6** | 77.2 | 76.8 | 76.1 |
| $l_\infty$ | 0.07 | 61.5 | 59.8 | 56.3 | 43.8 | 41.8 | **40.4** | 44.3 | 43.1 | 42.6 |
| | 0.09 | 33.2 | 46.7 | 41.0 | 16.5 | 14.2 | **12.8** | 16.2 | 14.8 | 14.4 |
| | 0.11 | 13.1 | 34.4 | 26.2 | 4.0 | 2.8 | **2.4** | 3.3 | 3.1 | **2.4** |
| | | CW | DF | LRA | PGD-1 | PGD-10 | PGD-100 | FAB-1 | FAB-10 | FAB-100 |
| | 0.5 | **92.6** | 93.6 | **92.6** | **92.6** | **92.6** | **92.6** | **92.6** | **92.6** | **92.6** |
| | 1 | 47.4 | 58.6 | 47.4 | 48.4 | 47.4 | **46.2** | 47.0 | 46.8 | **46.2** |
| $l_2$ | 1.5 | 8.8 | 19.8 | 7.8 | 9.8 | 8.8 | 8.2 | 7.8 | 7.2 | **7.0** |
| | 2 | 0.6 | 1.8 | **0.2** | 1.2 | 0.6 | 0.6 | **0.2** | **0.2** | **0.2** |
| | 2.5 | **0.0** | **0.0** | **0.0** | 0.6 | 0.2 | 0.2 | **0.0** | **0.0** | **0.0** |
| | | SparseFool | EAD | PGD-1 | PGD-10 | PGD-100 | FAB-1 | FAB-10 | FAB-100 |
| | 2 | 95.5 | 93.6 | 94.4 | 93.9 | 93.7 | 94.2 | 93.7 | **93.5** |
| | 4 | 88.9 | 76.7 | 79.8 | 77.5 | 76.9 | 80.2 | 76.6 | **75.2** |
| $l_1$ | 6 | 75.8 | 48.1 | 57.4 | 52.2 | 49.3 | 54.5 | 47.2 | **43.3** |
| | 8 | 60.3 | 26.6 | 46.7 | 36.3 | 31.6 | 31.3 | 25.3 | **22.4** |
| | 10 | 43.8 | 11.2 | 40.0 | 27.4 | 22.1 | 15.2 | 9.8 | **8.4** |

Table 6: Comparison of $l_\infty$-, $l_2$- and $l_1$-attacks on an $l_\infty$-robust model on MNIST. We report the accuracy on the test set if the attack is allowed to perturb the test points of $\epsilon$ in $l_p$-distance. The statistics are computed on the first 1000 points on the test set for $l_\infty$ and $l_1$, on 500 points for $l_2$.

**Robust accuracy of MNIST $l_\infty$-robust model**

| metric | $\epsilon$ | DF | DAA-1 | DAA-50 | PGD-1 | PGD-10 | PGD-100 | FAB-1 | FAB-10 | FAB-100 |
|---|---|---|---|---|---|---|---|---|---|---|
| | 0.2 | 95.2 | 94.6 | **93.7** | 95.0 | 94.2 | **93.7** | 94.6 | 94.4 | 93.9 |
| | 0.25 | 94.7 | 92.7 | **91.1** | 93.1 | 91.8 | 91.4 | 93.3 | 92.1 | 91.7 |
| $l_\infty$ | 0.3 | 93.9 | 89.5 | **87.2** | 91.3 | 88.3 | 87.6 | 91.2 | 89.2 | 88.5 |
| | 0.325 | 92.5 | 72.1 | **64.2** | 74.9 | 68.4 | 64.7 | 86.2 | 83.1 | 81.3 |
| | 0.35 | 89.8 | 19.7 | **11.7** | 32.1 | 19.3 | 13.8 | 48.7 | 32.0 | 23.8 |
| | | CW | DF | LRA | PGD-1 | PGD-10 | PGD-100 | FAB-1 | FAB-10 | FAB-100 |
| | 1 | 88.8 | 94.6 | 73.6 | 92.2 | 90.8 | 89.8 | 84.2 | 70.6 | **65.4** |
| | 1.5 | 77.6 | 93.0 | 25.8 | 86.0 | 81.2 | 77.0 | 47.0 | 20.6 | **12.2** |
| $l_2$ | 2 | 64.4 | 91.6 | 3.2 | 77.8 | 67.0 | 57.8 | 15.6 | 1.8 | **0.2** |
| | 2.5 | 53.8 | 89.6 | 0.4 | 68.2 | 49.6 | 36.4 | 3.8 | **0.0** | **0.0** |
| | 3 | 46.8 | 84.6 | **0.0** | 59.8 | 29.6 | 13.4 | 1.4 | **0.0** | **0.0** |
| | | SparseFool | EAD | PGD-1 | PGD-10 | PGD-100 | FAB-1 | FAB-10 | FAB-100 |
| | 2.5 | 96.8 | 92.2 | 94.1 | 93.7 | 93.6 | 90.1 | 74.3 | **56.9** |
| | 5 | 96.5 | 76.0 | 90.9 | 88.9 | 88.2 | 85.8 | 39.4 | **17.6** |
| $l_1$ | 7.5 | 96.4 | 49.5 | 85.2 | 81.4 | 79.0 | 82.6 | 19.8 | **5.0** |
| | 10 | 96.4 | 27.4 | 80.2 | 73.5 | 70.3 | 78.4 | 11.9 | **2.4** |
| | 12.5 | 96.4 | 14.6 | 74.9 | 65.6 | 58.7 | 74.5 | 7.7 | **0.5** |

Table 7: Comparison of $l_\infty$-, $l_2$- and $l_1$-attacks on an $l_2$-robust model on MNIST. We report the accuracy in percentage of the classifier on the test set if the attack is allowed to perturb the test points of $\epsilon$ in $l_p$-distance. The statistics are computed on the first 1000 points on the test set for $l_\infty$ and $l_1$, on 500 points for $l_2$.

**Robust accuracy of MNIST $l_2$-robust model**

| metric | $\epsilon$ | DF | DAA-1 | DAA-50 | PGD-1 | PGD-10 | PGD-100 | FAB-1 | FAB-10 | FAB-100 |
|---|---|---|---|---|---|---|---|---|---|---|
| $l_\infty$ | 0.05 | 96.7 | 96.4 | **96.3** | 96.4 | **96.3** | **96.3** | 96.4 | **96.3** | **96.3** |
| | 0.1 | 93.4 | 91.0 | **90.2** | 90.7 | 90.4 | **90.2** | 90.8 | 90.4 | 90.4 |
| | 0.15 | 86.4 | 74.3 | 72.3 | 74.6 | 73.2 | 72.4 | 74.0 | 72.3 | **72.0** |
| | 0.2 | 73.8 | 34.5 | 27.2 | 36.2 | 29.8 | 26.5 | 34.1 | 28.2 | **24.4** |
| | 0.25 | 55.1 | 1.5 | 0.9 | 2.6 | 1.5 | 1.0 | 1.9 | 0.9 | **0.8** |

| metric | $\epsilon$ | CW | DF | LRA | PGD-1 | PGD-10 | PGD-100 | FAB-1 | FAB-10 | FAB-100 |
|---|---|---|---|---|---|---|---|---|---|---|
| $l_2$ | 1 | **92.6** | 93.8 | **92.6** | 93.0 | 93.0 | 93.0 | **92.6** | **92.6** | **92.6** |
| | 1.5 | 84.8 | 87.2 | **83.4** | 83.8 | **83.4** | **83.4** | 83.8 | 83.6 | 83.6 |
| | 2 | 70.6 | 79.0 | 68.0 | 68.8 | 68.0 | **67.6** | 69.8 | 69.0 | 67.8 |
| | 2.5 | 46.4 | 67.4 | 41.6 | 45.6 | 40.4 | **37.6** | 45.6 | 41.8 | 39.2 |
| | 3 | 17.2 | 54.2 | 11.2 | 17.4 | 12.4 | **10.2** | 18.6 | 13.4 | 11.0 |

| metric | $\epsilon$ | SparseFool | EAD | PGD-1 | PGD-10 | PGD-100 | FAB-1 | FAB-10 | FAB-100 |
|---|---|---|---|---|---|---|---|---|---|
| $l_1$ | 5 | 94.9 | **89.8** | 90.3 | 90.2 | 90.2 | 90.5 | 90.2 | 90.0 |
| | 8.75 | 89.1 | **71.2** | 75.5 | 74.0 | 72.7 | 75.3 | 73.7 | 72.2 |
| | 12.5 | 81.0 | 45.9 | 61.1 | 57.5 | 54.9 | 55.6 | 49.2 | **45.7** |
| | 16.25 | 72.8 | **20.6** | 49.2 | 42.3 | 38.4 | 32.2 | 24.1 | 20.8 |
| | 20 | 60.8 | 8.3 | 41.4 | 29.6 | 23.2 | 15.2 | 9.4 | **7.7** |

Table 8: Comparison of $l_\infty$-, $l_2$- and $l_1$-attacks on a naturally trained model on CIFAR-10. We report the accuracy in percentage of the classifier on the test set if the attack is allowed to perturb the test points of $\epsilon$ in $l_p$-distance. The statistics are computed on the first 1000 points on the test set for $l_\infty$ and $l_1$, on 500 points for $l_2$.

**Robust accuracy of CIFAR-10 plain model**

| metric | $\epsilon$ | DF | DAA-1 | DAA-50 | PGD-1 | PGD-10 | PGD-100 | FAB-1 | FAB-10 | FAB-100 |
|---|---|---|---|---|---|---|---|---|---|---|
| $l_\infty$ | $1/255$ | 62.6 | 65.7 | 64.1 | 56.1 | 55.8 | **55.6** | 56.5 | 55.9 | 55.7 |
| | $1.5/255$ | 49.3 | 63.2 | 60.8 | 38.9 | 37.9 | **37.4** | 38.5 | 37.7 | **37.4** |
| | $2/255$ | 37.3 | 62.4 | 58.5 | 24.3 | 23.3 | 22.9 | 23.4 | 21.9 | **21.2** |
| | $2.5/255$ | 26.4 | 61.2 | 56.3 | 16.2 | 14.8 | 14.0 | 13.2 | 12.0 | **11.8** |
| | $3/255$ | 19.0 | 60.2 | 54.4 | 10.7 | 9.2 | 8.6 | 7.4 | 5.8 | **5.4** |

| metric | $\epsilon$ | CW | DF | LRA | PGD-1 | PGD-10 | PGD-100 | FAB-1 | FAB-10 | FAB-100 |
|---|---|---|---|---|---|---|---|---|---|---|
| $l_2$ | 0.1 | 69.4 | 72.2 | 69.0 | 68.4 | **67.6** | **67.6** | 68.4 | 68.4 | 68.4 |
| | 0.15 | 55.4 | 62.6 | 55.0 | 54.6 | **53.8** | **53.8** | 54.6 | 54.0 | **53.8** |
| | 0.2 | 43.4 | 51.2 | 43.4 | 43.8 | 42.8 | 42.0 | 42.4 | 42.0 | **41.8** |
| | 0.3 | 21.6 | 33.8 | 22.0 | 24.8 | 24.2 | 23.6 | 21.6 | 20.8 | **20.6** |
| | 0.4 | 9.4 | 20.8 | 9.8 | 18.2 | 16.2 | 15.4 | 9.6 | 8.2 | **8.0** |

| metric | $\epsilon$ | SparseFool | EAD | PGD-1 | PGD-10 | PGD-100 | FAB-1 | FAB-10 | FAB-100 |
|---|---|---|---|---|---|---|---|---|---|
| $l_1$ | 2 | 72.1 | 54.7 | 54.9 | 54.4 | 53.9 | 55.5 | 52.2 | **50.8** |
| | 4 | 58.6 | 24.1 | 30.0 | 29.1 | 28.9 | 30.7 | 25.1 | **22.4** |
| | 6 | 45.6 | 8.9 | 18.8 | 18.6 | 18.4 | 17.0 | 10.5 | **8.1** |
| | 8 | 34.3 | 3.0 | 14.2 | 14.1 | 14.0 | 7.8 | 3.8 | **2.5** |
| | 10 | 27.2 | **0.7** | 12.9 | 12.5 | 12.3 | 4.7 | 1.5 | 1.0 |

Table 9: Comparison of $l_\infty$-, $l_2$- and $l_1$-attacks on an $l_\infty$-robust model on CIFAR-10. We report the accuracy in percentage of the classifier on the test set if the attack is allowed to perturb the test points of $\epsilon$ in $l_p$-distance. The statistics are computed on the first 1000 points on the test set for $l_\infty$ and $l_1$, on 500 points for $l_2$.

**Robust accuracy of CIFAR-10 $l_\infty$-robust model**

| metric | $\epsilon$ | DF | DAA-1 | DAA-50 | PGD-1 | PGD-10 | PGD-100 | FAB-1 | FAB-10 | FAB-100 |
|---|---|---|---|---|---|---|---|---|---|---|
| $l_\infty$ | $2/255$ | 66.8 | 66.9 | 66.3 | **65.5** | **65.5** | **65.5** | 65.8 | 65.8 | 65.7 |
| | $4/255$ | 53.2 | 63.8 | 61.4 | 49.8 | 49.3 | 49.0 | 49.2 | 49.1 | **48.9** |
| | $6/255$ | 42.9 | 63.1 | 58.4 | 38.0 | 36.9 | 36.6 | 35.4 | 34.7 | **34.6** |
| | $8/255$ | 32.9 | 61.2 | 56.3 | 30.5 | 30.0 | 29.6 | 23.8 | 23.5 | **23.3** |
| | $10/255$ | 24.5 | 59.8 | 54.1 | 25.8 | 23.7 | 22.4 | 15.4 | 14.7 | **14.4** |

| metric | $\epsilon$ | CW | DF | LRA | PGD-1 | PGD-10 | PGD-100 | FAB-1 | FAB-10 | FAB-100 |
|---|---|---|---|---|---|---|---|---|---|---|
| $l_2$ | 0.25 | 64.6 | 67.0 | **64.4** | **64.4** | **64.4** | **64.4** | 64.8 | 64.6 | **64.4** |
| | 0.5 | 48.4 | 53.0 | 48.8 | 49.0 | 48.4 | **48.0** | 48.4 | 48.4 | 48.2 |
| | 0.75 | 33.4 | 41.4 | 33.4 | 39.0 | 38.2 | 37.4 | 33.6 | 33.2 | **33.0** |
| | 1 | 22.8 | 32.6 | 22.8 | 35.0 | 34.4 | 33.8 | 22.2 | 21.6 | **21.4** |
| | 1.25 | 12.0 | 24.2 | 13.0 | 34.6 | 34.2 | 33.2 | 12.2 | **11.2** | **11.2** |

| metric | $\epsilon$ | SparseFool | EAD | PGD-1 | PGD-10 | PGD-100 | FAB-1 | FAB-10 | FAB-100 |
|---|---|---|---|---|---|---|---|---|---|
| $l_1$ | 5 | 57.8 | **36.8** | 47.3 | 46.6 | 46.2 | 43.1 | 39.9 | 37.9 |
| | 8.75 | 44.7 | **19.2** | 37.4 | 37.0 | 36.8 | 25.7 | 22.5 | 20.2 |
| | 12.5 | 34.9 | **7.1** | 34.0 | 33.9 | 33.9 | 13.7 | 10.9 | 8.7 |
| | 16.25 | 27.6 | **3.0** | 33.3 | 33.2 | 33.1 | 7.1 | 4.3 | 3.5 |
| | 20 | 20.2 | **0.9** | 32.9 | 32.8 | 32.8 | 3.8 | 1.7 | 1.3 |

Table 10: Comparison of $l_\infty$-, $l_2$- and $l_1$-attacks on an $l_2$-robust model on CIFAR-10. We report the accuracy in percentage of the classifier on the test set if the attack is allowed to perturb the test points of $\epsilon$ in $l_p$-distance. The statistics are computed on the first 1000 points on the test set for $l_\infty$ and $l_1$, on 500 points for $l_2$.

**Robust accuracy of CIFAR-10 $l_2$-robust model**

| metric | $\epsilon$ | DF | DAA-1 | DAA-50 | PGD-1 | PGD-10 | PGD-100 | FAB-1 | FAB-10 | FAB-100 |
|---|---|---|---|---|---|---|---|---|---|---|
| $l_\infty$ | $2/255$ | 64.1 | 67.2 | 66.3 | 62.6 | 62.5 | **62.4** | 62.7 | 62.6 | 62.6 |
| | $4/255$ | 49.0 | 65.0 | 62.8 | 45.3 | 45.0 | 44.9 | 44.4 | **44.2** | **44.2** |
| | $6/255$ | 36.9 | 64.2 | 60.8 | 32.9 | 31.6 | 31.1 | 27.2 | 26.8 | **26.7** |
| | $8/255$ | 25.8 | 62.3 | 58.0 | 25.7 | 24.9 | 23.9 | 14.8 | 14.1 | **13.8** |
| | $10/255$ | 17.6 | 61.9 | 54.8 | 21.9 | 19.8 | 18.6 | 8.6 | 8.0 | **7.9** |

| metric | $\epsilon$ | CW | DF | LRA | PGD-1 | PGD-10 | PGD-100 | FAB-1 | FAB-10 | FAB-100 |
|---|---|---|---|---|---|---|---|---|---|---|
| $l_2$ | 0.25 | 66.0 | 67.0 | **65.6** | 65.8 | **65.6** | **65.6** | **65.6** | **65.6** | **65.6** |
| | 0.5 | 48.2 | 53.8 | **47.8** | 49.6 | 48.8 | 48.8 | 48.4 | 48.2 | 48.0 |
| | 0.75 | 32.6 | 42.2 | 32.4 | 38.4 | 37.2 | 36.4 | 32.8 | 32.4 | **32.2** |
| | 1 | 21.6 | 30.0 | 21.6 | 35.4 | 33.6 | 33.0 | 21.8 | 21.4 | **21.0** |
| | 1.25 | **11.4** | 22.4 | 12.4 | 34.2 | 31.6 | 31.2 | 12.2 | 12.2 | **11.4** |

| metric | $\epsilon$ | SparseFool | EAD | PGD-1 | PGD-10 | PGD-100 | FAB-1 | FAB-10 | FAB-100 |
|---|---|---|---|---|---|---|---|---|---|
| $l_1$ | 3 | 69.5 | **62.2** | 64.5 | 64.4 | 64.4 | 63.4 | 63.2 | 63.0 |
| | 6 | 61.6 | **45.5** | 51.9 | 51.9 | 51.8 | 48.6 | 47.2 | 45.6 |
| | 9 | 53.1 | **27.7** | 42.8 | 42.5 | 42.5 | 33.9 | 30.6 | 28.8 |
| | 12 | 44.4 | 17.9 | 38.5 | 38.3 | 38.0 | 23.8 | 19.8 | **17.3** |
| | 15 | 37.0 | **10.4** | 35.8 | 35.8 | 35.4 | 16.0 | 12.4 | 11.2 |

Table 11: Comparison of $l_\infty$-, $l_2$- and $l_1$-attacks on a naturally trained model on Restricted ImageNet. We report the accuracy in percentage of the classifier on the test set if the attack is allowed to perturb the test points of $\epsilon$ in $l_p$-distance. The statistics are computed on the first 500 points of the test set.

**Robust accuracy of Restricted ImageNet plain model**

| metric | $\epsilon$ | DF | DAA-1 | DAA-10 | PGD-1 | PGD-10 | FAB-1 | FAB-10 |
|---|---|---|---|---|---|---|---|---|
| $l_\infty$ | $0.25/255$ | 76.6 | 74.8 | 74.8 | 74.8 | **74.6** | 75.2 | 75.2 |
| | $0.5/255$ | 52.0 | 51.8 | 48.2 | 38.2 | **37.8** | 39.6 | 39.6 |
| | $0.75/255$ | 26.8 | 46.0 | 41.0 | **12.2** | **12.2** | 14.2 | 14.2 |
| | $1/255$ | 11.2 | 43.2 | 39.4 | 3.8 | 3.8 | **3.6** | **3.6** |
| | $1.25/255$ | 5.0 | 41.2 | 38.2 | **1.0** | **1.0** | 1.2 | **1.0** |

| metric | $\epsilon$ | | DF | PGD-1 | PGD-10 | FAB-1 | FAB-10 |
|---|---|---|---|---|---|---|---|
| $l_2$ | 0.2 | | 80.2 | **76.0** | **76.0** | 77.0 | 76.8 |
| | 0.4 | | 58.4 | 40.8 | **40.6** | 43.0 | 42.2 |
| | 0.6 | | 33.8 | 15.4 | **14.8** | 19.0 | 18.2 |
| | 0.8 | | 18.8 | **4.0** | **4.0** | 4.6 | 4.4 |
| | 1 | | 8.6 | 1.6 | 1.6 | 1.2 | **1.0** |

| metric | $\epsilon$ | | SparseFool | PGD-1 | PGD-5 | FAB-1 | FAB-5 |
|---|---|---|---|---|---|---|---|
| $l_1$ | 5 | | 88.6 | 81.8 | 81.8 | 79.6 | **78.0** |
| | 16 | | 80.0 | 45.2 | 45.2 | 46.8 | **40.0** |
| | 27 | | 70.6 | 17.8 | **17.4** | 25.6 | 19.8 |
| | 38 | | 65.0 | 6.2 | **6.0** | 13.6 | 7.0 |
| | 49 | | 55.4 | **2.2** | **2.2** | 6.8 | 3.8 |

Table 12: Comparison of $l_\infty$-, $l_2$- and $l_1$-attacks on an $l_\infty$-robust model on Restricted ImageNet. We report the accuracy in percentage of the classifier on the test set if the attack is allowed to perturb the test points of $\epsilon$ in $l_p$-distance. The statistics are computed on the first 500 points of the test set.

**Robust accuracy of Restricted ImageNet $l_\infty$-robust model**

| metric | $\epsilon$ | DF | DAA-1 | DAA-10 | PGD-1 | PGD-10 | FAB-1 | FAB-10 |
|---|---|---|---|---|---|---|---|---|
| $l_\infty$ | $2/255$ | 75.8 | 75.0 | 75.0 | **74.6** | **74.6** | 75.2 | 75.2 |
| | $4/255$ | 53.0 | 46.2 | 46.2 | **45.4** | **45.4** | 47.4 | 47.4 |
| | $6/255$ | 32.4 | 24.6 | 23.8 | **19.4** | **19.4** | 21.2 | 21.0 |
| | $8/255$ | 19.4 | 17.0 | 14.6 | **6.2** | **6.2** | 6.8 | 6.8 |
| | $10/255$ | 10.8 | 12.8 | 11.6 | 1.0 | **0.8** | 1.2 | 1.2 |

| metric | $\epsilon$ | | DF | PGD-1 | PGD-10 | FAB-1 | FAB-10 |
|---|---|---|---|---|---|---|---|
| $l_2$ | 1 | | 79.4 | **76.6** | **76.6** | 77.0 | 76.8 |
| | 2 | | 65.0 | 46.8 | **46.2** | 49.8 | 49.2 |
| | 3 | | 46.8 | 22.4 | **21.4** | 24.4 | 23.8 |
| | 4 | | 32.8 | 9.0 | **8.6** | 10.8 | 10.6 |
| | 5 | | 20.4 | 3.0 | **2.8** | 3.2 | 3.2 |

| metric | $\epsilon$ | | SparseFool | PGD-1 | PGD-5 | FAB-1 | FAB-5 |
|---|---|---|---|---|---|---|---|
| $l_1$ | 15 | | 81.8 | 69.8 | 69.8 | 69.6 | **68.2** |
| | 25 | | 76.4 | 58.6 | 58.2 | 56.2 | **53.6** |
| | 40 | | 71.4 | 41.0 | 41.0 | 41.2 | **37.6** |
| | 60 | | 63.2 | 28.8 | 28.8 | 28.2 | **23.8** |
| | 100 | | 49.2 | 12.0 | 11.6 | 14.6 | **11.2** |

Table 13: Comparison of $l_\infty$-, $l_2$- and $l_1$-attacks on an $l_2$-robust model on Restricted ImageNet. We report the accuracy in percentage of the classifier on the test set if the attack is allowed to perturb the test points of $\epsilon$ in $l_p$-distance. The statistics are computed on the first 500 points of the test set.

**Robust accuracy of Restricted ImageNet $l_2$-robust model**

| metric | $\epsilon$ | DF | DAA-1 | DAA-10 | PGD-1 | PGD-10 | FAB-1 | FAB-10 |
|---|---|---|---|---|---|---|---|---|
| | $^2/_{255}$ | 74.4 | **73.0** | **73.0** | **73.0** | **73.0** | 73.8 | 73.8 |
| | $^4/_{255}$ | 49.0 | 39.6 | 39.2 | **37.6** | **37.6** | 39.6 | 39.4 |
| $l_\infty$ | $^6/_{255}$ | 27.4 | 22.6 | 21.0 | **13.2** | **13.2** | 15.0 | 15.0 |
| | $^8/_{255}$ | 13.8 | 18.6 | 16.8 | 3.6 | 3.6 | 3.6 | **3.2** |
| | $^{10}/_{255}$ | 6.6 | 15.6 | 13.8 | **0.4** | **0.4** | 0.8 | 0.8 |
| | | | DF | PGD-1 | PGD-50 | FAB-1 | FAB-10 | |
| | 2 | | 74.2 | **71.8** | **71.8** | 72.8 | 72.8 | |
| | 3 | | 61.6 | 51.4 | **51.0** | 52.4 | 52.4 | |
| $l_2$ | 4 | | 45.6 | 31.0 | **30.8** | 34.4 | 33.8 | |
| | 5 | | 34.6 | **20.4** | **20.4** | 22.6 | 21.8 | |
| | 6 | | 25.2 | **9.6** | **9.6** | 11.8 | 11.6 | |
| | | | SparseFool | PGD-1 | PGD-5 | FAB-1 | FAB-5 | |
| | 50 | | 85.4 | 81.0 | 81.0 | 78.6 | **78.4** | |
| | 100 | | 79.6 | 63.8 | 63.6 | 60.8 | **59.0** | |
| $l_1$ | 150 | | 74.4 | 48.4 | 48.4 | 45.2 | **42.2** | |
| | 200 | | 68.6 | 32.2 | 32.2 | 31.0 | **29.0** | |
| | 250 | | 60.0 | 23.0 | 22.4 | 22.8 | **20.2** | |

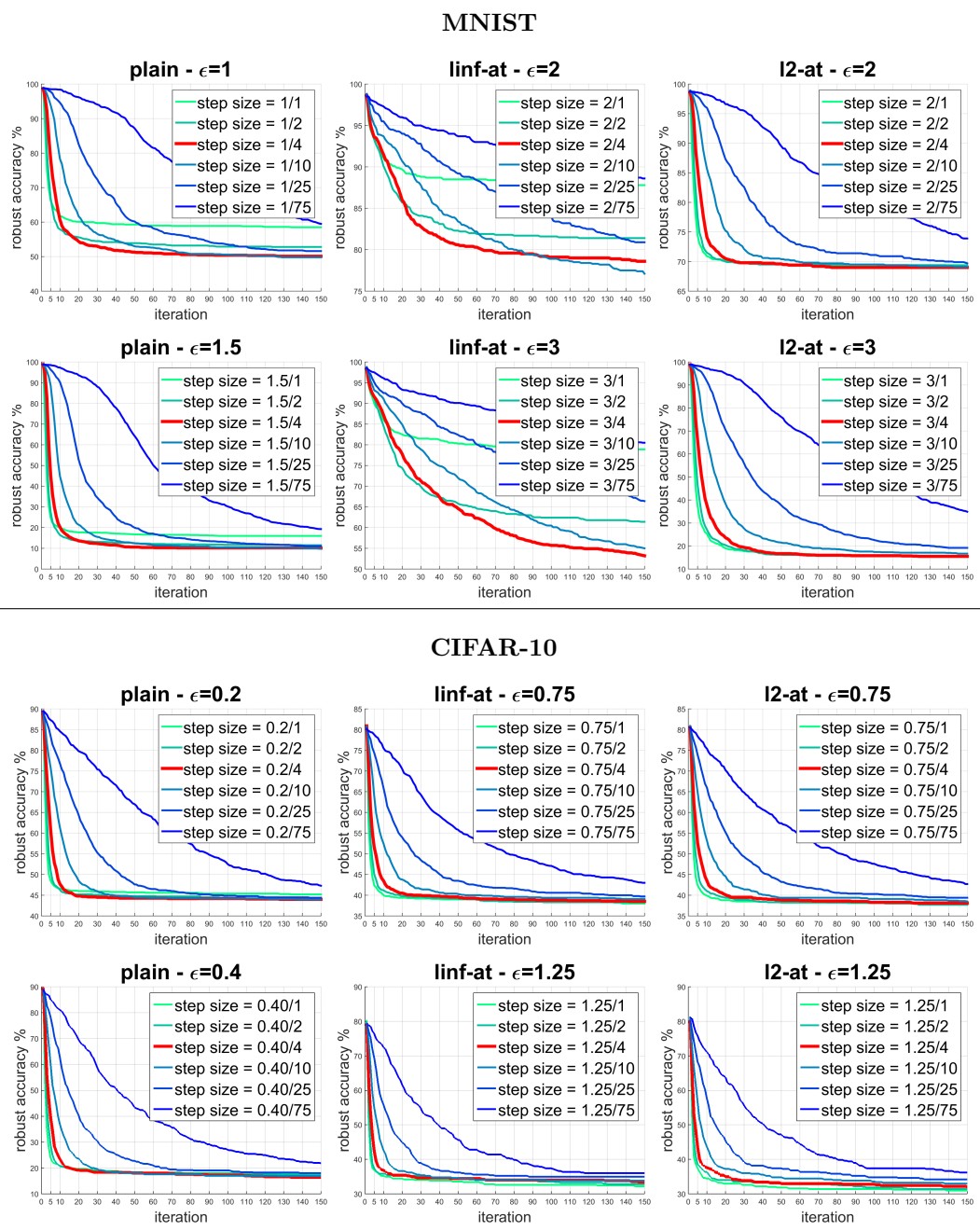

Figure 3: We plot for different step sizes of PGD robust accuracy over iterations. In red the step size we used in the experiments of Section 3. We clearly see that our chosen step-size is on average the best one. The models used are those trained on MNIST (top row) and CIFAR-10 (bottom row).

# B    Analysis of the attacks

## B.1    Choice of the step size of PGD

We here show the performance of PGD wrt $l_2$ on MNIST and CIFAR-10 under different choices of the step size. In particular we focus here on the largest $\epsilon$ and the middle $\epsilon$ values chosen in the evaluation where the different stepsize choices have the largest impact. We report the robust accuracy for at each of the 150 iterations. We test step sizes $\epsilon/t$ for $t \in \{1, 2, 4, 10, 25, 75\}$. For each step size we run the attack 10 times with random initialization and show the run which achieves the lowest robust accuracy after 150 iterations. Note however that the behaviour of different runs varies minimally. In Figure 3 we show the results for the three models for MNIST and CIFAR-10 for two different choices of $\epsilon$ used in Section 3, with step size decreasing the blue becoming darker, while our chosen step size, that is $\epsilon/4$, is highlighted in red. We see that it achieves in all the models best or close to best robust accuracy and is clearly the best on average.

## B.2    Evolution across iteration

We here want to compare the evolution of the robust accuracy across the iterations of a single run of PGD and FAB, that is PGD-1 and FAB-1 from Tables 5 to 13. Since PGD performs 1 forward and 1 backward pass for each iteration and FAB 2 forward passes and 1 backward pass, we rescale the robust accuracy so to compare the two methods when they have exploited the same number of passes of the network. Then 300 passes correspond to 150 iterations of PGD and to 100 of FAB. In Figures 4, 5 and 6 we show the evolution of robust accuracy for the different dataset, models and threat models ($l_\infty$, $l_2$ and $l_1$), computed at the threshold $\epsilon$ median among the five used in Tables 5 to 13.

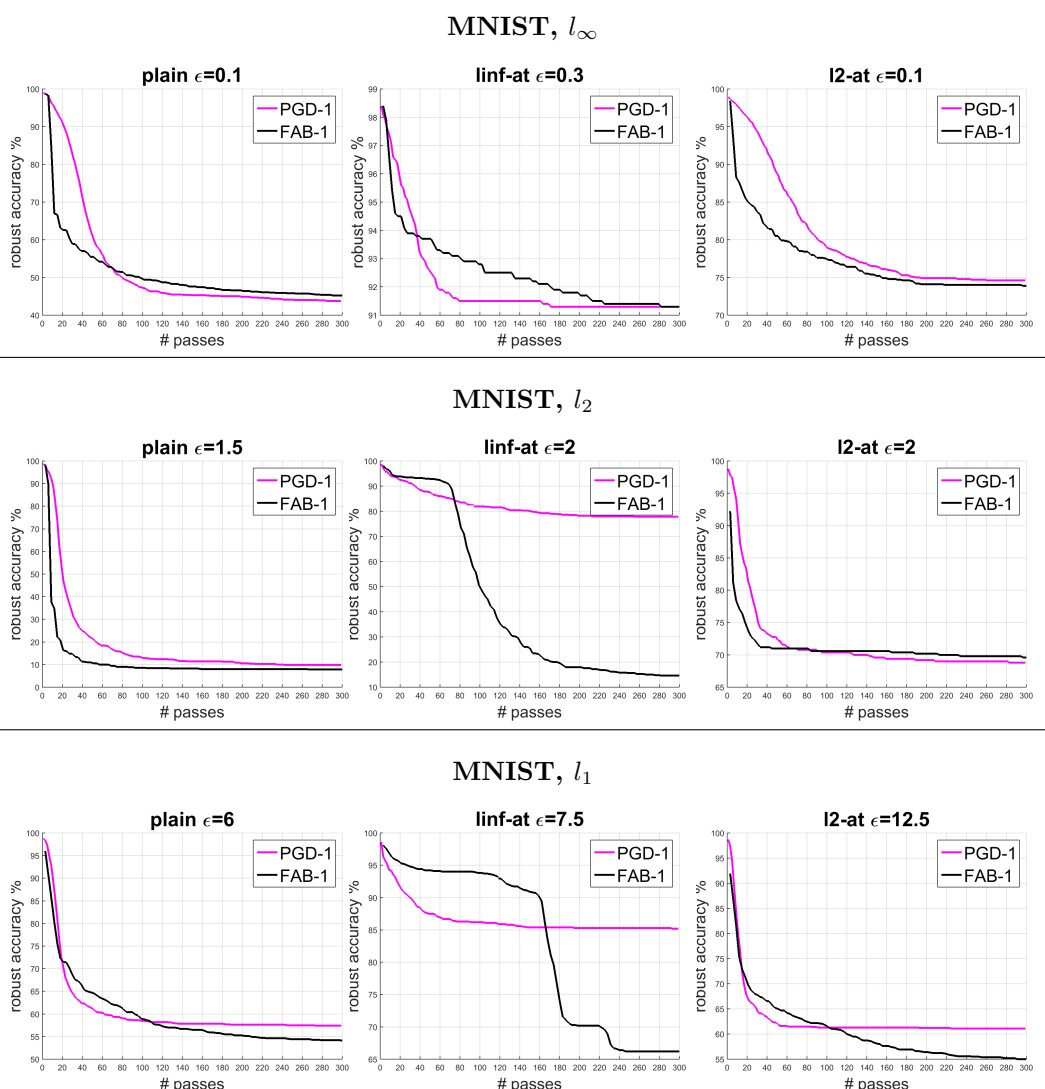

Figure 4: Evolution of accuracy across iterations on MNIST. We compare the robust accuracy of PGD-1 (magenta) and FAB-1 (black) as a function of the employed forward/backward passes in the algorithm (one iteration of PGD corresponds to 2 passes, one iteration of FAB corresponds to 3 passes). Models: plain in the first column, $l_\infty$-at in the second and $l_2$-at in the third. Threat models: $l_\infty$ in the first row, $l_2$ in the second and $l_1$ in the third. The thresholds $\epsilon$ used can be read above the plots. Note that the result of FAB-1 on $l_\infty$-at wrt $l_1$ does not match that in Table 6 due to a typo in the statistics in the table.

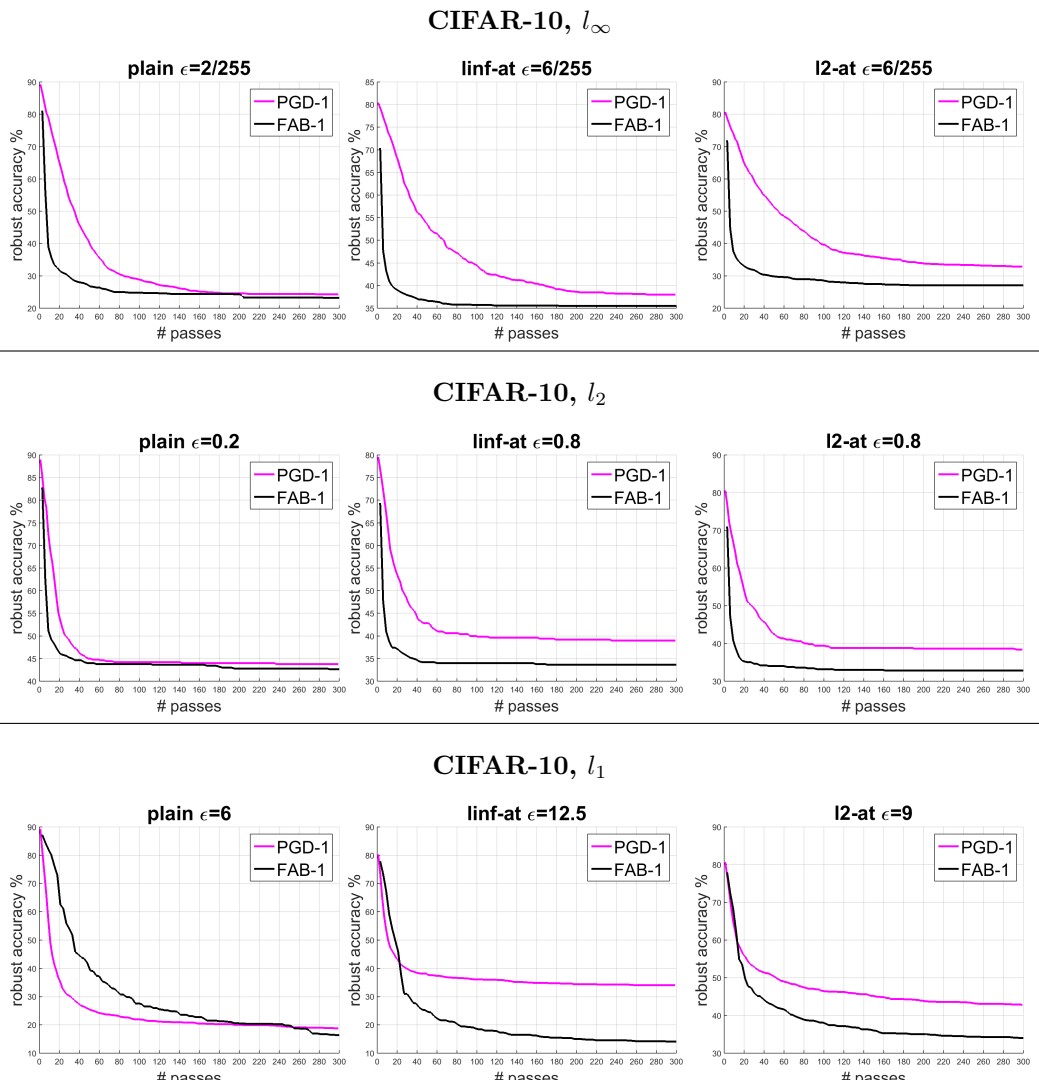

Figure 5: Evolution of accuracy across iterations on CIFAR-10. We compare the robust accuracy of PGD-1 (magenta) and FAB-1 (black) as a function of the employed forward/backward passes in the algorithm (one iteration of PGD corresponds to 2 passes, one iteration of FAB corresponds to 3 passes). Models: plain in the first column, $l_\infty$-at in the second and $l_2$-at in the third. Threat models: $l_\infty$ in the first row, $l_2$ in the second and $l_1$ in the third. The thresholds $\epsilon$ used can be read above the plots.

**Restricted ImageNet, $l_\infty$**

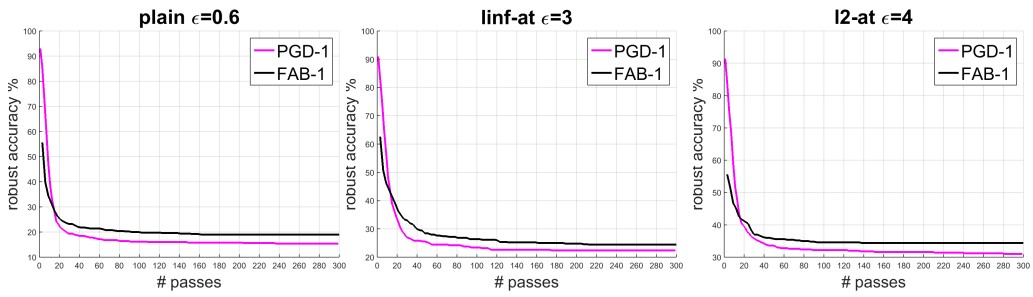

**Restricted ImageNet, $l_2$**

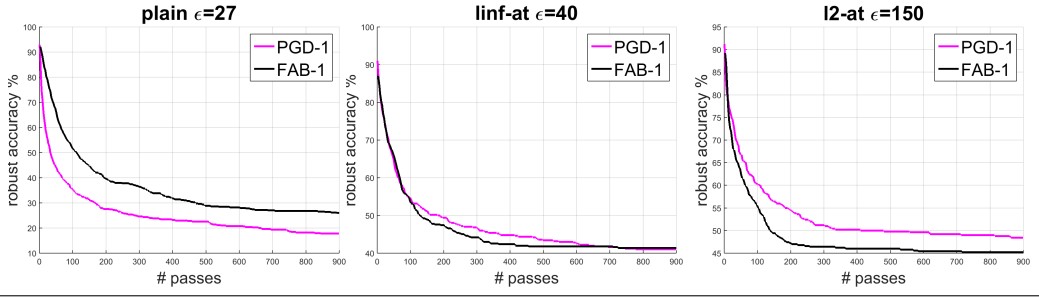

**Restricted ImageNet, $l_1$**

Figure 6: Evolution of robust accuracy across iterations on Restricted ImageNet. We compare the robust accuracy of PGD-1 (magenta) and FAB-1 (black) as a function of the employed forward/backward passes in the algorithm (one iteration of PGD corresponds to 2 passes, one iteration of FAB corresponds to 3 passes). Models: plain in the first column, $l_\infty$-at in the second and $l_2$-at in the third. Threat models: $l_\infty$ in the first row, $l_2$ in the second and $l_1$ in the third. The thresholds $\epsilon$ used can be read above the plots.

