# OpenReview forum: "Minimally distorted Adversarial Examples with a Fast Adaptive Boundary Attack"
_ICLR.cc/2020/Conference — Reject_

### Official Review · AnonReviewer3 · 2019-10-20
**Official Blind Review #3**

**Rating:** 6

**Review:**

The paper studies the problem of the white-box attack of neural network-based classifiers, with an emphasis on the "minimal distortion solution": The new input that changes the labeling output of the network with the minimal distance (l1, l2, l_inf) with respect to a given input.

The main intuition of the algorithm is to do a local linear approximation of the network at the current point (which is the Taylor expansion up to the gradient term). After that, the algorithm identifies a class (output coordinate) with the minimal "margin to gradient norm ratio", i.e. the total movement in gradient direction to change the labeling function in that coordinate, within this linear approximation. The algorithm solves the subproblem of minimizing a linear function inside lp ball as the critical routine.

Overall, the notion of finding the minimal distortion attacker as opposed to finding the best attacker inside a fixed distortion ball is quite interesting to me. The main concern for me about this paper is the comparison to other methods such as PGD. As far as I know, these attackers DO NOT explicitly minimize the distortion, thus it is quite believable that these models do not identify the minimal distortion solution (rather it will more likely to find a solution that lies in the boundary since it would be the easiest way to attack). However, for the proposed algorithm in this paper, the algorithm is explicitly minimizing the distance to the given input (x_orig in their language).


I would like to see more implementation details of the other algorithms, for example, what is the performance if we add an additional regularizer as the distance of the current attacker to the given input to PGD. So far, the paper lacks solid proof of the usefulness of this particular algorithm. (In particular the justification for solving the local linear system instead of doing a gradient descent step).

After Rebuttal: I have read the authors' responses and acknowledge the sensibility of the statement. I apologize for the earlier misunderstanding and higher the score accordingly.


**Experience Assessment:**

I have published in this field for several years.

**Review Assessment: Checking Correctness Of Derivations And Theory:**

I assessed the sensibility of the derivations and theory.

**Review Assessment: Checking Correctness Of Experiments:**

I assessed the sensibility of the experiments.

**Review Assessment: Thoroughness In Paper Reading:**

I read the paper thoroughly.

---

> ### Author Response · Authors · 2019-11-13
> **Answer to reviewer 3**
>
> We thank the reviewer for the comments. We address below the questions.
>
> "The main concern for me about this paper is the comparison to other methods such as PGD. As far as I know, these attackers DO NOT explicitly minimize the distortion, thus it is quite believable that these models do not identify the minimal distortion solution (rather it will more likely to find a solution that lies in the boundary since it would be the easiest way to attack). However, for the proposed algorithm in this paper, the algorithm is explicitly minimizing the distance to the given input (x_orig in their language)."
>
> It is right that PGD is not aiming at the minimal adversarial change. However, please note that we evaluate our models exactly in a way which is fair to PGD, that means robust test error for the threat model of an l_p-ball of fixed epsilon. This
> means we are just evaluating if the classifier does change its decision inside this l_p-ball or not but we are not comparing the size of the distortions. In particular, we therefore run PDG for each choice of epsilon again and indeed the found adversarial examples are typically located at the boundary of the l_p-ball
> but since we just check if the decision changes this is counted in the same way even if the adversarial distortion found by FAB has smaller norm.
> In the appendix A.4 (Table 4) we additionally report the average minimal distortions found by other methods which also aim at minimal distortions e.g. EAD, CW, LRA and DF. In this case we don't report the results of PGD and DAA exactly because they don't aim at minimizing the norm and it would be an improper comparison.
>
> We hope that the reviewer readjusts his/her score after this major misunderstanding has been clarified.
>
> "I would like to see more implementation details of the other algorithms, for example, what is the performance if we add an additional regularizer as the distance of the current attacker to the given input to PGD. So far, the paper lacks solid proof of the usefulness of this particular algorithm. (In particular the justification for solving the local linear system instead of doing a gradient descent step)."
>
> Note that for the other algorithms we mainly take them (see Section 3, paragraph Attacks) as implemented in cleverhans (Papernot et al. 2017) and foolbox (Rauber et al 2017) or directly the code provided by the authors (DAA, LRA). Only SparseFool and PGD were implemented by us. Note that it is not necessary to add a
> regularization term to PGD as we are not aiming at minimal adversarial distortions but just to change the class and therefore maximizing the cross-entropy loss
> of the correct class as done in PGD is perfectly aligned with this goal.
>
> Regarding our FAB attack, a clear advantage of projecting on the approximated decision hyperplane over doing a gradient step is that the projection does not need to fix a step size, but rather, in practice, adaptively chooses the optimal step size. While our method has also some hyperparameters they generalize across
> models, datasets and threat models (l_\infty, l_2, l_1).
> Second, as noticed by Reviewer 2, FAB does not suffer from gradient obfuscation, as can be seen for the results wrt l_2 and l_1 on the l_\infty-adversarially trained model of (Madry et al, 2018) in Table 6. Third, it is fast and at the same time produces high quality adversarial examples (in Section 3 we show it is competitive or outperforms attacks specialized in just one of the three norms,
> see also the additional experiments in Appendix B.2). Since it minimizes the norm of the perturbations, it provides quickly a complete overview of the robustness of a classifier at every threshold.

---

### Official Review · AnonReviewer2 · 2019-10-21
**Official Blind Review #2**

**Rating:** 6

**Review:**

The authors propose a new gradient-based method (FAB) for constructing adversarial perturbations for deep neural networks. At a high level, the method repeatedly estimates the decision boundary based on the linearization of the classifier at a given point and projects to the closest "misclassified" example based on that estimation (similar to DeepFool). The authors build on this idea, proposing several improvements and evaluate their attack empirically against a variety of models.

I found the proposed method quite interesting and intuitive. All the improvements made to the core method are well-motivated and clearly explained, while the ablation experiments are relatively thorough.

However, I did find the presentation of experimental evidence quite misleading.

Specifically, reporting mean accuracy over models, datasets, and epsilon constraints in Table 2 does not give the full picture. Going through the appendix tables, we can see the following:
-- The step size used for PGD is quite large---eps/4 for the L2 case---which is quite uncommon when using 150 iterations. Based on prior work and my own personal experience, a step size of 2 * eps / #steps (i.e., eps / 75) would seem more suitable. I wonder if this is the reason for PGD performing worse than FAB for large epsilon values on CIFAR10. The authors mention that they chose this parameter using grid search but do not provide concrete details.
-- The adversarially trained MNIST model of Madry et al. 2018 learns to use thresholding filters as the first layer (observed in the original paper). This causes issues for most gradient-based methods (e.g., PGD performs worse than the decision-based attack of Brendel et al. 2018, also observed in other prior work). While it is encouraging that FAB is robust to such gradient obfuscation, this is arguably not the ideal setting to compare gradient based methods (especially when averaging performance over models).
-- For MNIST and Restricted IN, PGD performs comparably or even better than FAB (modulo larger epsilon values for which the large step size used could be an issue for PGD and the Linf-trained model with the thresholding filters).
-- For the L1-norm setting, EAD performs similarly or better compared to FAB (again modulo the Linf-trained model).
Based on these observations, I am not fully convinced that FAB outperforms PGD (for L2 and Linf) and EAD (for L1) by as much as Table 2 suggests.

Moreover, the runtime comparison performed in not exactly fair:
-- It is not clear how many restarts where included in the runtime of PGD. Its runtime should be in the same ballpark as FAB but the time reported is ~20x higher.
-- PGD is known to produce quite accurate estimates when run with much fewer (say 15) steps. Thus in order to make a fair comparison one would also need to look at the entire #steps vs robust accuracy curve to get a better picture of the efficiency of these two methods. Choosing an arbitrary number of steps for each method is not very enlightening.
-- It is not necessary to run PGD 5 times to evaluate the robust accuracy at 5 thresholds. One can perform binary search for each input in order to find the smallest epsilon for which a misclassification can be found. This will result in at most 3 (sometimes 2) evaluations per point (instead of 5).

Despite these shortcomings of the experimental evaluation, I still believe that the paper has merit. After all, the method is clean and well-motivated,  performs comparably to the best of PGD and EAD in a variety of settings, and is robust to a certain degree of gradient masking. In that sense, it could potentially be a valuable contribution and could be of interest to a subset of the adversarial ML community.

In the sense, while my initial stance is to recommend (weak) rejection, I would be open to increasing my score and recommending (weak) acceptance should my concerns be addressed.

UPDATE: I appreciate the response and the additional experiments performed by the authors. The authors have addressed my concerns in their response. I am increasing my score to a weak accept.

One thing that would be nice to add in the next version of the manuscript is a note inviting the reader to consider the appendix tables since average robust accuracy can be inconclusive.

**Experience Assessment:**

I have published in this field for several years.

**Review Assessment: Checking Correctness Of Derivations And Theory:**

I carefully checked the derivations and theory.

**Review Assessment: Checking Correctness Of Experiments:**

I carefully checked the experiments.

**Review Assessment: Thoroughness In Paper Reading:**

I read the paper thoroughly.

---

> ### Author Response · Authors · 2019-11-13
> **Answer to reviewer 2**
>
> We thank the reviewer for the detailed comments. We are sorry that our evaluation was perceived as misleading. Since we ran extensive experiments on 3 datasets, 3 models, 3 norms, with 5 thresholds and 8 competitors plus our attack, we needed a concise summary, hence the statistics of Tables 1/2. In fact we are not aware of any other attack paper with such an extensive evaluation and comparison. In particular we tried hard to run each attack with optimal parameters.
>
> "The step size used for PGD is quite large---eps/4 for the L2 case---which is quite uncommon when using 150 iterations. Based on prior work and my own personal experience, a step size of 2 * eps / #steps (i.e., eps / 75) is suitable..."
>
> We added in Appendix B.1 the performance of PGD with different step sizes epsilon/t for t\in\{1, 2, 4, 10, 25, 75\}  for the l_2 attack on MNIST/ CIFAR-10.  We report in Figure 6 for each step size the the robust accuracy over iterations (best run out of 10 for each step size) for different epsilon. Our chosen stepsize of \epsilon/4 for PGD achieves best/close to best results in all cases and is best on average. We chose this stepsize by optimizing average PGD performance over all models for MNIST and CIFAR-10 trying out 8 different stepsizes. Thus we took care that we run PGD with the best possible parameters.
>
> "While it is encouraging that FAB is robust to such gradient obfuscation, this is arguably not the ideal setting to compare gradient based methods (especially when averaging performance over models)."
> " ...EAD performs similarly or better compared to FAB (again modulo the Linf-trained model)."
>
> It is a favorable property that FAB is robust to gradient obfuscation which is also a clear advantage over PGD and other attacks affected by gradient obfuscation. However, we see the point of the reviewer that this  could affect the overall statistics in Table 1 for
> l_2 and l_1. Thus we report additionally the statistics of Table 1 for l_2 and l_1 without considering the l_infty-adversarially trained model of (Madry et al, 2018). FAB still outperforms the competitors, with the only exception of "# best" for l_1 (13 of EAD vs 12 of FAB).
>
> The difference between FAB and EAD is small when not considering Madry's model l_infty model but since FAB does not suffer from gradient obfuscation, while EAD does so, we think it is fair to say that FAB outperforms EAD. While FAB does not always outperform PGD for l_infty, FAB is the best attack in the summary statistics, in particular it is never far away from the best result. Please note that even FAB-10 outperforms PGD-100
>
> "Based on these observations, I am not fully convinced that FAB outperforms PGD (for L2 and Linf) and EAD (for L1) by as much as Table 2 suggests."
>
> We hope that the new evaluation and illustration of our step size choice for PGD convince the reviewer of the opposite.
>
> "It is not clear how many restarts where included in the runtime of PGD. Its runtime should be in the same ballpark as FAB but the time reported is ~20x higher."
>
> In the corresponding paragraph we write: ``"if not specified otherwise, it includes all the restarts". However, to improve clarity we added the number of restarts. PGD-100 (100 restarts, 150 iterations) and FAB-100  (100 restarts, 100 iterations) are comparable as they do both 300 forward/backward passed for one run. PGD-100 takes 3820s on MNIST for 5 thresholds (764s for one threshold), whereas FAB-100 takes 1613s.  In theory PGD-100 should take the same amount of time as FAB-100 (300 forward/backward passes per restart). The difference is most likely a suboptimal implementation of the gradients of FAB and we are currently trying to fix this.
>
> "PGD is known to produce quite accurate estimates when run with much fewer (say 15) steps. Thus in order to make a fair comparison ... the entire #steps vs robust accuracy curve ..."
>
> We agree but then this would yield 135 plots (5 thresholds, 3 datasets, 3 models, 3 norms). We have also the problem that the time per step is not the same for all the methods and typically varies with the hardware and implementation. We have added a comparison of FAB-1 and PGD-1 in Appendix B.2 wrt to the number of forward/backward passes (1 iter. PGD: 2 passes, 1 iter. FAB: 3 passes) . We cannot confirm that PGD yields always good results already within 15 steps (=30 passes). In the 27  reported cases (Figure 4,5,6 in the Appendix) FAB-1 is better in 18 out of 27 cases after 20 passes. However, both methods sometimes which require the full number of passes to get good results.
>
> "It is not necessary to run PGD 5 times to evaluate the robust accuracy at 5 thresholds. One can perform binary search .... This will result in at most 3... evaluations per point ..."
>
> Thanks for pointing this out. We report runtime for PGD in this way in the final version. However, we still think that for a detailed robustness evaluation a method like FAB evaluating the robustness curve in one run is of advantage.

---

### Official Review · AnonReviewer1 · 2019-10-24
**Official Blind Review #1**

**Rating:** 6

**Review:**

Authors extend deepFool by adding extra steps and constraints to find closer points to the source image as the adversarial image. They both project onto the decision boundary. Deepfool does and adhoc clipping to keep the pixel values in (0,1) but the new proposed method respects the constraints during the steps. Also during the steps they combine projection of last step result and original image to keep it closer to the original image. Moreover, at the end of the optimization they perform extra search steps to get closer to the original image. Also they add random restarts. Rather than considering the original image, they randomly choose an image in the half ballpark of the total delta.

According to the results in fig.2 the backward steps has the highest impact in comparison to deepfool. But mixing with original projection always helps a little and random restarts help a little too. Without the backward steps there is almost no gain from mixing the projections.

Considering the full results in the appendix, the results are mixed with no obvious advantage in comparison to PGD specially.

**Experience Assessment:**

I have published one or two papers in this area.

**Review Assessment: Checking Correctness Of Derivations And Theory:**

I assessed the sensibility of the derivations and theory.

**Review Assessment: Checking Correctness Of Experiments:**

I assessed the sensibility of the experiments.

**Review Assessment: Thoroughness In Paper Reading:**

I made a quick assessment of this paper.

---

> ### Author Response · Authors · 2019-11-13
> **Answer to reviewer 1**
>
> We thank the reviewer for the helpful comments.
>
> "According to the results in fig.2 the backward steps has the highest impact in comparison to deepfool. But mixing with original projection always helps a little and random restarts help a little too. Without the backward steps there is almost no gain from mixing the projections."
>
> Our projection (alpha=0.1, no restarts) compared to a ``Deep-Fool with backward step'' (alpha=0, no restarts) is always better and improves robust accuracy by more than 10% in the area between eps=0.35 and eps=0.38. We think this is significant. The overall method with 100 restarts (alpha=0.1, 100 restarts) compared to
> ``Deep-Fool with backward step'' (alpha=0, 100 restarts) is again always better and improves robust accuracy by 5-10% on a wide range of epsilon values. This makes the difference between FAB-attack which produces state-of-the-art results or some attack which is ok.
>
> "Considering the full results in the appendix, the results are mixed with no obvious advantage in comparison to PGD specially"
>
> We think there are quite some quantitative and qualitative advantages of FAB-attack over PGD which we highlight below. Nevertheless, it is clear that the computation of a minimal attack or robust accuracy is a non-convex optimization problem and thus it is unlikely that there will be ever something like the best attack algorithm (unless you solve the mixed-integer program directly). Our experiments are honest, extensive (many different models (normal, robust) and data-sets) and contain no cherry picking.
>
> Highlights of FAB-attack (compared to PGD and other methods)
> i) FAB-attack achieves for all norms on average the best robust accuracy, is closest to the best on average and has no dramatic failure cases (maximum difference to the best). This is still the case when we take out the models where gradient obfuscation is a problem and PGD fails (please see answer to Reviewer 2)
>
> ii) Compared to PGD, FAB-attack requires no step-size. For PGD at least for each norm,  but potentially for each model, one has to tune the step-size parameter of PDG for optimal performance. In particular, in our experience for attacking new defense strategies, one has to carefully tune the stepsize parameter of PGD. Note that we have selected the optimal parameters for PGD via a grid-search for every norm separately by taking the one achieving best performance on average on MNIST and CIFAR-10. Please see attack details in A.2 and B.1 and the answer to reviewer 2. In contrast, for FAB-attack all parameters are constant across all threat models on MNIST and CIFAR-10, with little adjustment for Restricted ImageNet.
>
> iii) as noticed by Reviewer 2, FAB does not suffer from gradient obfuscation, as can be seen for the results wrt l_2 and l_1 on the l_infty-adversarially trained model of (Madry et al, 2018) in Table 6.
>
> iv) FAB aims at the minimal adversarial perturbation and thus provides with one run a complete robustness evaluation, which is different for PGD where one evaluates the robustness at a fixed threshold (even though we agree with reviewer 2 that in order to evaluate different thresholds for PGD it is not necessary to run the attack for all data points again).
>
> v) FAB has achieved for one very competitive public challenge the lowest reported robust test accuracy for a robust model on MNIST. It has also obtained on two other public challenges the lowest robust accuracy but has been outperformed since then by a new attack scheme. As these challenges are running some time already, all major attacks have been tried there. Unfortunately, we cannot be more concrete here without violating the anonymity policy.

---

### Decision · Program_Chairs · 2019-12-19

**Decision:**

Reject

**Comment:**

This work presents a method for generating an (approximately) minimal adversarial perturbation for neural networks. During the discussion period, the AC raised additional concerns that were not originally addressed by the reviewers. The method is an iterative first order method for solving constrained optimization problems, however when considered as a new first order optimization method the contribution seems minimal. Most of the additions are rather straightforward---e.g. using a line search at each step to determine the optimal step size---and the reported gains over PGD are unconvincing. PGD can be considered as a "universal" first order optimizer [1], as such we should be careful that the reported gains are substantial and not just a question of tuning. Given that using a line search at each step increases the computational cost by a multiplicative factor, the comparison with PGD should take this into account.

The AC notes several plots in the Appendix show PGD having better performance (particularly on restricted Imagenet), and for others there remain questions on how PGD is tuned (for example the CIFAR-10 plots in Figure 5). One of two things explains the discrepancies in Figure 5: either PGD is finding a worse local optimum than FAB, or PGD has not converged to a local optimum. There needs to be provided experiments to rule out the second possibility, as this is evidence that PGD is not being tuned properly. Some standard things to check are the step size and number of steps. Additionally, enforcing a constant step size after projection is an easy way to improve the performance of PGD. For example, if the gradient of the loss is approximately equal to the normal vector of the constraint, then proj(x_i+ lambda * g) ~ x_i will result in an effective step size that is too low to make progress.

Finally, it is unclear what practical use there is for a method that finds an approximately minimum norm perturbation. There are no provable guarantees so this cannot be used for certification. Additionally, in order to properly assess the security and reliability of ML systems, it is necessary to consider larger visual distortions, occlusions, and corruptions (such as the ones in [2]) as these will actually be encountered in practice.

1. https://arxiv.org/pdf/1706.06083.pdf
2. https://arxiv.org/abs/1807.01697